# Plant conversions and abatement technologies cannot prevent stranding of power plant assets in 2 °C scenarios

Yangsiyu Lu [1], Francois Cohen [2✉], Stephen M. Smith [3] & Alexander Pfeiffer[3]

Continued fossil fuel development puts existing assets at risk of exceeding the capacity compatible with limiting global warming below 2 °C. However, it has been argued that plant conversions and new abatement technologies may allow for a smoother transition. We quantify the impact of future technology availability on the need for fossil fuel power plants to be stranded, i.e. decommissioned or underused. Even with carbon capture and storage (CCS) and bioenergy widely deployed in the future, a total of 267 PWh electricity generation (ten times global electricity production in 2018) may still be stranded. Coal-to-gas conversions could prevent 10–30 PWh of stranded generation. CCS retrofits, combined with biomass co-firing, could prevent 33–68 PWh. In contrast, lack of deployment of CCS or bioenergy could increase stranding by 69 or 45 percent respectively. Stranding risks remain under optimistic technology assumptions and even more so if CCS and bioenergy are not deployed at scale.

[1] Smith School of Enterprise and the Environment, University of Oxford; Institute for New Economic Thinking, University of Oxford; Global Development Policy Center, Boston University, Boston, MA, USA. [2] Department of Economics, Chair of Energy Sustainability and Barcelona Institute of Economics, University of Barcelona, Carrer de John Maynard Keynes, 1, 11, 08034 Barcelona, Spain. [3] Smith School of Enterprise and the Environment, University of Oxford, Oxford, UK.
✉email: francois.cohen@ub.edu

Because fossil fuel capacity has drastically expanded in the last decade[1], there are now strong concerns that the existing stock of fossil-fuel power plants, if used as planned, could exhaust global carbon budgets consistent with the aims of the Paris Agreement[2–4]. However, the energy industry has argued that technological solutions, especially energy efficiency, carbon capture and storage (CCS), bioenergy, natural offsets and switching from coal to gas, could substantially reduce the carbon footprint of energy production from fossil fuels and allow production to continue for longer without jeopardising the Paris Agreement[5–10]. These arguments come at a critical moment since most countries are currently deciding how fast they should decarbonise their economies. Coal phase-out plans have just started to be adopted in several countries with very different timelines.

So far, no study has estimated the effect of technology deployment and plant conversions on the stranding of existing assets, leaving some uncertainty in the required pace for the energy transition in the power sector. Compared to prior assessments that have focused on calculating committed emissions from existing infrastructure and the impact of climate policy on stranded assets[11–17], this paper accounts for the possible response of the industry towards developing technological solutions when facing the risk of assets being stranded. Not accounting for this response may overestimate the threat that existing fossil fuel infrastructure could pose to reaching the 2 °C targets of the Paris Agreement. Furthermore, looking at differences in assumptions regarding technology development allows us to reconcile differences in results across different models of long-term energy generation.

In this work, we assess the potential for new technologies and plant conversions to reduce the need for assets to be stranded, i.e. underused or retired early to ensure that global warming remains below 2 °C[18]. We follow Pfeiffer et al.[15] and define stranded assets in the power sector in terms of stranded generation: the reduction in electricity generation from fossil fuels arising from the necessary underuse and early decommission of power plants consistent with a 2 °C scenario. We firstly estimate stranded generation in all technologies deployed scenarios and find that a total of 267 PWh electricity generation (ten times global electricity production in 2018) may be at risk of stranding even if CCS and bioenergy are

widely deployed in the 21st century. We then assess the effect of technologies (CCS, bioenergy, alternative electricity supply and energy efficiency) and plant conversions (from coal to gas or biomass) on the amount of stranded generation. We find coal-to-gas plant conversions could prevent 10–30 PWh of stranded generation. CCS retrofits, combined with biomass co-firing, could prevent 33–68 PWh. Nevertheless, insufficient deployment of CCS or bioenergy could increase stranding by 69% or 45%, respectively. Our results suggest that, even in the presence of a strong industry response to develop low-carbon and negative emission technologies or convert current and planned assets to be less carbon-intensive, the expected amount of required stranding would remain substantial.

## Results

**Estimates of electricity generation and stranded generation.** We estimate the amount of stranded generation by comparing the amount of electricity forecast to be produced by power plants currently operating, under construction and planned, with energy production pathways consistent with a 2 °C target at the end of the century. Firstly, we compile generator-level power plant data globally by merging S&P Global Platt's World Electric Power Plants Database, Global Coal Plant Tracker, and World Resource Institute's Global Power Plant Database. We use the compiled dataset to estimate the future electricity generation from the power plants that are currently operating and in the pipeline.

Our estimates for the potential electricity generation from operating and in-the-pipeline fossil fuel power plants are provided in Fig. 1. Using baseline assumptions for utilisation rates and lifetimes (see baseline assumption details in the Methods section), we estimate that these power plants could produce 540 PWh in total between 2021 and 2100. As shown in Fig. 1a, about two-thirds of this electricity would come from coal-fired power plants, and the other third from gas-fired power plants. Only a negligible amount would come from oil-fired power plants. The currently operating plants would produce slightly more electricity than the plants in the pipeline. Figure 1b shows that around 60% of this electricity would come from Asia, and 20% from the OECD. The remaining 20% is shared between the Middle East, Africa, Latin America and Reforming Economies.

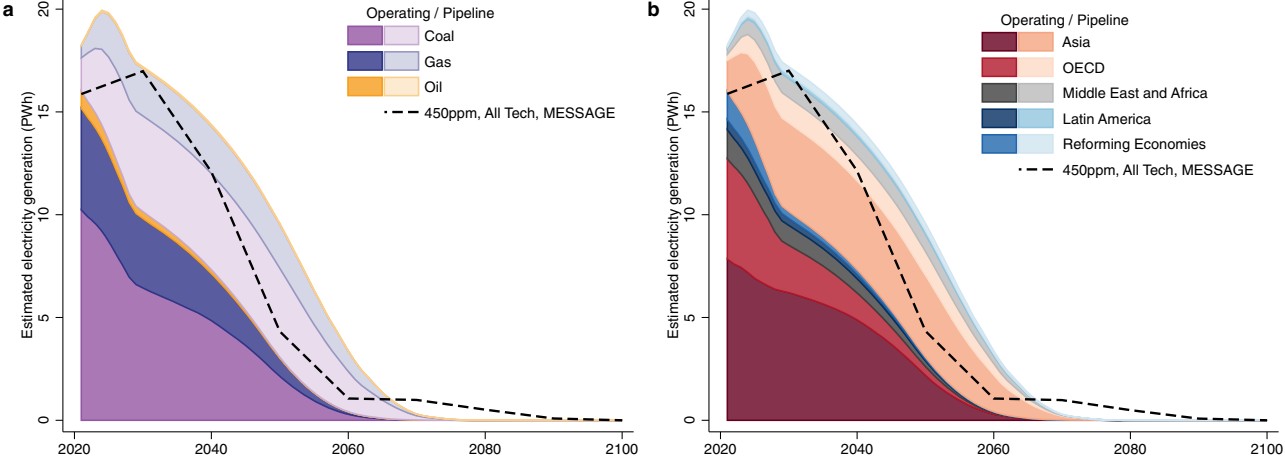

**Fig. 1 Estimated electricity generation between 2021 and 2100, by fuel (a) and by region (b).** We estimate the amount of future electricity that can be generated from currently operating power plants and those in the pipeline. Power units are assumed to be operating at the same utilisation rates as those forecasted in the IEA Stated Policy Scenarios (with a breakdown by category of power units described in Supplementary Table 2) until the end of their lifetime. Darker shading indicates the expected generation from currently operating plants, while lighter shading indicates the expected generation from in-the-pipeline plants. The black dashed line is an example of the electricity that could be produced based on one scenario from one Integrated Assessment Model, specifically the 450 ppm, all technologies deployed scenario obtained from the MESSAGE model. All the electricity generation shown in this figure is from fossil fuel plants without carbon capture and storage.

| Table 1 Technology settings in AMPERE scenarios. | | |
|---|---|---|
| **Technology dimensions** | **All technologies deployed scenarios** | **With one technology insufficiently deployed scenarios** |
| Carbon capture and storage | CCS is fully available. | CCS never becomes available, including for both fossil fuel and bio-based plants. |
| Nuclear | Nuclear energy is fully available. | No new investments into nuclear power after 2020; existing plants are fully phased out over their lifetime. |
| Solar and wind | Advanced* techno-economic assumptions for solar and wind technologies. | Limited contribution of solar and wind to 20% of total power generation, reflecting potential implementation barriers of renewable energy at high penetration rates. |
| Bioenergy | Total global bio-energy supply shall top out at the level generated endogenously by each model. | Total global bio-energy supply for all sectors from purpose-grown crops, residues and municipal solid waste shall be limited to 100 EJ/ year as primary energy. |
| Other setting Energy intensity | Energy intensity improves at historical rates. | A combination of efficiency measures and behavioural changes leads roughly to a 50% increase of the energy intensity improvement rate compared to historical rates. |

Source: AMPERE Working Package 2 model comparison study protocol and Riahi et al.[20]. *It is left to the modeller's choice what is being considered advanced.

Secondly, we compare the forecast of electricity generation of Fig. 1 with scenarios consistent with a 2 °C target. The scenarios are extracted from the AMPERE database, which gathers model outputs from different integrated assessment models (IAMs) and assumes that GHG emissions stabilise at 450 ppm by the end of the century. This is widely consistent with achieving the 2 °C targets in the literature[19,20]. The general characteristics, socio-economic drivers and key energy technology assumptions of these IAMs are presented in Supplementary Table 1.

Stranded generation is simply computed as the difference between the electricity generation allowed in the climate scenarios, and the electricity generation that operating and in-the-pipeline power plants could produce at baseline utilisation rates and lifetimes (see baseline assumption details in the Methods section).

An illustration is provided in Fig. 1. The black dashed lines correspond to one of the scenarios available in the AMPERE Working Package 2 (WP2) database. This scenario assumes that all low-carbon technologies available in the model can be deployed. In this case, electricity generation from currently operating plants is roughly within the example's modelled boundary. Nevertheless, adding the electricity generation from in-the-pipeline plants exceeds the scenario's modelled boundary, signalling the stranding risk. Other pathways that do not assume the availability of all technologies may lead to higher levels of stranded generation. Furthermore, stranding risk varies by fuel type, with stranding risk for coal-fired power plants much stronger than for other fuels. Separate pathways by fuel type for the same example are provided in Supplementary Fig. 1.

We are particularly interested in the AMPERE scenarios that assume the four technologies are fully deployed: (1) CCS; (2) bioenergy; (3) solar and wind and (4) nuclear energy, which we refer to as the all technologies deployed scenarios. Table 1 provides more detail on the definition of the technologies fully deployed scenarios compared to other scenarios. In a nutshell, no pre-set constraint is imposed on the function of technology deployment. In other scenarios, limits are usually imposed, whether the potential is limited (e.g. bioenergy limited to 100EJ per year or solar and wind are limited to 20% of total power generation) or the technology is not allowed (e.g. CCS is not available and no new investments in nuclear power after 2020). All technologies deployed scenarios also assume that energy intensity continues to reduce every year at historical rates. Supplementary Fig. 2 displays the global electricity generation by technology in all technologies deployed scenarios of each IAM.

Figure 2 provides our estimates of stranded generation for the all technologies deployed scenarios included in the AMPERE WP2 database. Results are provided for seven IAMs (GCAM, IMA-CLIM, IMAGE, MESSAGE, POLES, REMIND and WITCH), and for operating and in-the-pipeline plants separately. We furthermore provide the average amount of stranded generation across these seven models. Estimates of stranded generation are displayed at global level, for five regions (Asia, OECD, Middle East and Africa, Latin America and Reforming Economies) and the top two countries by volume of stranded generation (China and India).

Taking the average across the IAMs, Fig. 2 shows that about 267 PWh electricity, which is about 50% of the future electricity generated from currently operating and in-the-pipeline fossil fuel plants, would have to be stranded to achieve the objective of stabilising GHG concentrations at 450 ppm, under the assumption that CCS, bioenergy, solar, wind and nuclear energy are fully deployed. More than 50% of the stranded generation globally would be located in Asia, outweighing the total of stranded generation in the other four regions. China and India would face averages of 104 and 37 PWh of stranded generation, respectively.

About two-thirds of the required stranding is linked to electricity that will be generated in plants that have not been built yet but are in the pipeline. A large amount of stranding could therefore be avoided by stopping the construction of the plants that are currently planned or under construction. In Supplementary Fig. 3, we break down our estimates of stranded generation by fuel type. Coal-fired power plants make most of the stranded generation (82% across all IAMs), followed by gas-fired power plants contributing 16% in the across model mean. Due to their limited capacity, oil-fired plants only constitute a very small share of stranded generation.

The results presented in Figs. 1, 2 are estimated using baseline utilisation rates and lifetimes. We conduct a sensitivity analysis on the utilisation rates and lifetimes assumptions in supplementary results (see Supplementary Fig. 4). Higher utilisation rates and longer lifetime assumptions would lead to more future electricity generation and therefore more stranded generation. However, even though the share of stranded generation increases with longer lifetimes and higher utilisation rates, it remains close to about 50% of future electricity generation in most cases.

The amount of stranded generation varies significantly across different IAMs. This variation arises from the substantial differences in structures and assumptions used by IAMs[20,21]. The existence of these differences is directly observable in Fig. 2, where the amount of stranded generation globally is lowest in

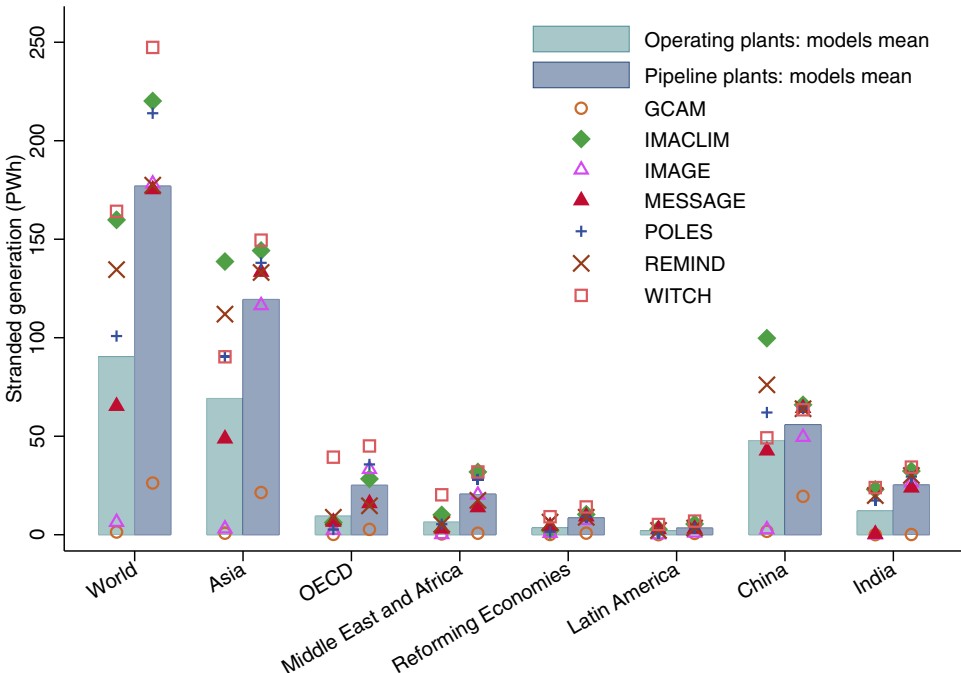

**Fig. 2 Estimated stranded generation in all technologies deployed scenarios (2021–2100).** Total stranded generation of fossil fuel power plants (including coal-, gas- and oil-fired) from 2021 to 2100 in all technologies deployed scenarios of AMPERE consistent with greenhouse gas concentrations stabilise at 450 ppm. Stranded generation are computed by fuel and then aggregated. We assume no conversion of the currently operating and in-the-pipeline fossil-fuel power plants to be equipped with carbon capture and storage, and no conversion of the currently operating and in-the-pipeline coal-fired power plants to bioenergy or gas. Scatter points represent the estimation from individual integrated assessment model and bars show the means across all models.

GCAM (about 28 PWh), and highest in WITCH (nearly 411 PWh). GCAM is the only IAM with very low levels of stranded generation, with all the other models entailing at least 180 PWh of stranded generation. This is due to high levels of bioenergy deployment in GCAM scenarios: electricity produced by biomass with CCS (BECCS) could reach as much as 180 EJ per year in 2100 in GCAM, while other models only forecast production of 6–60 EJ per year (see Supplementary Fig. 2). The biomass potential assumed in GCAM is higher than the widely accepted global sustainable biomass potential of 100 EJ per year[22]. More electricity supplied from biomass with CCS allows more electricity to be generated from fossil fuels and thus leads to lower stranded generation.

We have compared our AMPERE-derived results to those based on the IPCC SR1.5 database[23]. The IPCC SR1.5 database includes 1.5 as well as 2 °C scenarios. We estimate the amount of stranded generation with this database in Supplementary Fig. 5. The average amount of stranded generation across all models in 2 °C scenarios of IPCC SR1.5 is higher by 12% compared to the amount in Fig. 2. This is in part because AMPERE scenarios in Fig. 2 assume that all technologies capable of reducing stranded generation have been fully deployed, while 2 °C scenarios in IPCC SR1.5 have varying assumptions on technology development. The average amount of stranded generation across all models in the 1.5 °C scenarios of IPCC SR1.5 is about 17% higher than in the 2 °C scenarios of IPCC SR1.5.

The rest of this paper is based on the scenarios available in AMPERE, because they allow for the estimation of the effect of technology availability on stranded generation through the pairwise comparison of scenarios with and without a technology (as illustrated in Table 1).

**Impact of plant conversions.** Our analysis above suggests high levels of stranded generation under all IAMs apart from GCAM.

These estimates may be overly restrictive, however, because they do not consider that power plants could adapt to the risk of stranded generation through conversions in order to use lower-carbon technologies. In Fig. 3, we allow for power plants to be converted and estimate the impact that this would have on stranded generation (compared to Fig. 2). We consider three conversion options: (1) conversion from coal to gas; (2) the installation of CCS in existing plants and in those currently in the pipeline and (3) the use of biomass in coal-fired power plants.

Coal power plants can be modified to operate only on natural gas (conversion); to fire either coal or natural gas (dual fuel), or to fire both coal and natural gas at the same time (co-firing)[24–28]. Converting a coal-fired boiler to gas requires adding new equipment, such as gas igniters, scanners, piping and valves. It also requires a modification of burner management and combustion control systems, an adjustment of pressure-part through the convection pass and a layup of coal and ash handling equipment[26,29]. So far, coal-to-gas projects have been undertaken in more than 80 coal-fired power plants in the US between 2011 and 2019[30], representing about 5% of total US coal-fired power capacity. Other countries such as Canada[29] and United Kingdom[31] also have several coal-to-gas projects at different stages of development.

In Fig. 3a, we estimate the change of stranded generation when considering potential conversions from coal to gas. We consider that coal-fired power plants located in countries that also include operating gas-fired power plants are suitable for conversion since they have access to gas infrastructure and supplies. We then assume that between 5 and 20% of them could be converted. The 5% corresponds to the US conversion percentage between 2011 and 2019. Twenty per cent is an optimistic upper bound based on the possibility that gas is co-fired with coal. In that regard, Domeshek and Burtraw[32] have suggested that US coal-fired power plants could adopt a 20% gas co-firing standard. Because these figures of 5–20% are derived from the US experience, they

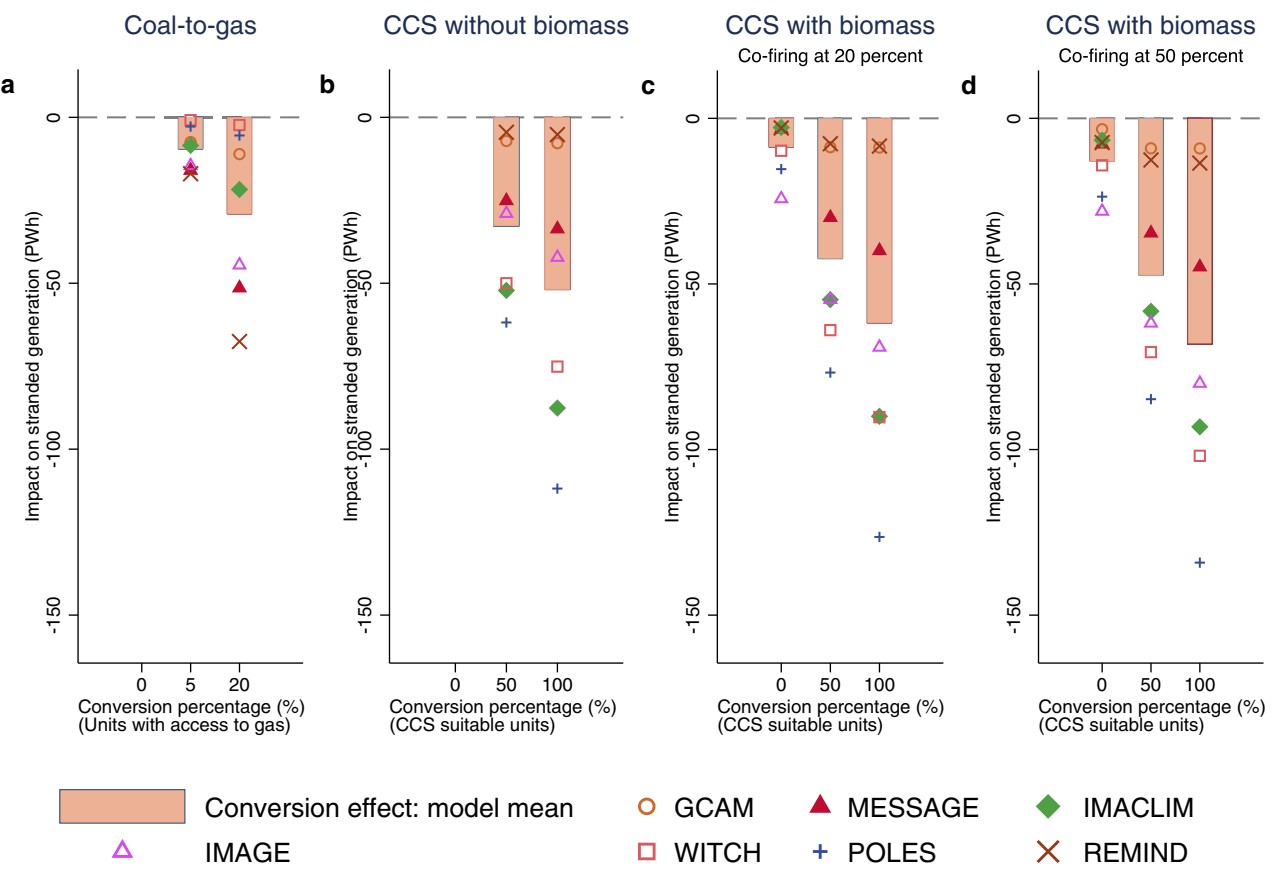

**Fig. 3 Impact of plant conversions on global stranded generation. a** Coal-to-gas, **b** carbon capture and storage (CCS) without biomass, **c** CCS with biomass co-firing at 20%, **d** CCS with biomass co-firing at 50%. Scatter points represent the estimation from individual models and bars show the model mean. The conversion percentages correspond to the share of coal-fired power plants converted to gas. Those of **b**–**d** correspond to the share of CCS suitable plants converted to CCS. Biomass co-firing ratios vary from 0 in b to 50% in **d**.

are likely to constitute upper bounds for what could happen globally. In the US, natural gas is relatively cheap because of the deployment of shale gas and the significant reduction in gas prices that followed. We furthermore limit possible plant conversions so that the amount of electricity produced from converted plants remains equal to or below the electricity produced from gas in the IAMs. Beforehand, we have subtracted the amount already covered by existing gas-fired power plants. In this way, our projections for the impact of plant conversions take into account future forecasts for the use of gas in power generation globally.

Results in Fig. 3a suggest that coal-to-gas conversions may reduce stranded generation by 10–30 PWh on average across all models. When we assume a maximum conversion potential of 5% of coal-fired power plants, on average 2.8% of coal-to-gas conversion occurs in our projections due to the limits imposed by IAMs regarding the use of coal and gas in power generation. When we allow for up to 20% of conversions, 10.6% of coal-fired generation is converted on average due to the limits imposed by the IAMs.

In Fig. 3b, we look at CCS adoption from operating and in-the-pipeline fossil fuel power plants. Caldecott et al.[33] assess that existing generation units are likely to be suitable for CCS installation if they have a capacity above 100 MW, are less than 20 years old, emit less than 1000 g CO2/kWh and are located within 40 km of geological areas suitable for CCS. Using these criteria, we find that around 24% of operating and in-the-pipeline fossil fuel power units would be suitable for CCS conversion. In Fig. 3b,

we show the impact of CCS conversion on stranded generation if either half of those suitable capacities, or all of them, would be equipped with CCS. We furthermore make sure that CCS installation in existing and in-the-pipeline power plants coincides with the take-up of CCS in the IAMs. Therefore, we bound our estimates of electricity generated from converted plants with CCS to be equal or inferior to the total volume of electricity generated from fossil fuels with CCS in the IAMs. The results displayed in Fig. 3b imply that stranded generation can be reduced by 33–52 PWh thanks to the future installation of CCS in operating and in-the-pipeline plants. Reductions are most pronounced in the IAMs in which we found the largest amounts of stranded generation (i.e. POLES, IMACLIM and WITCH).

In Fig. 3c, d, we consider that coal-fired power plants (with and without CCS) could be adapted to co-fire biomass. So far, more than 150 power plants have fired coal along with biomass, with the majority located in northern Europe and the US[34]. However, in the IAMs, 95% of future electricity generation with biomass comes from plants equipped with CCS. Our calculations, therefore, have to consider the co-firing of biomass in some plants that are not equipped with CCS, and the co-firing of biomass in plants that would be retrofitted with CCS. Figure 3c, d consider three scenarios of CCS retrofits: no plants are converted to CCS; 50% of CCS suitable capacities are converted to CCS, and all CCS suitable capacities are converted to CCS. The last two assumptions are the same as in Fig. 3b. Furthermore, we allow coal plants to co-fire with biomass. Co-firing ratios can vary. The IEA and IRENA[35] estimate that a 20% co-firing ratio is feasible in

most cases, while a 50% co-firing ratio is technically achievable. Figure 3c assumes a co-firing ratio of 20% and Fig. 3d has a co-firing ratio of 50%.

In both figures, the coal-to-biomass potential is restricted to the availability of sustainable biomass feedstock within each IAM. Co-firing without CCS cannot exceed the electricity generated from biomass without CCS in the IAMs, after accounting for the electricity that would already be generated from existing biomass plants. Furthermore, biomass co-firing has to be equal or inferior to either 20 or 50% of the total energy generated from converted plants: there cannot be more biomass co-fired than the amount of coal allows for.

In the absence of any CCS conversion, Fig. 3c suggests that biomass co-firing could reduce the amount of stranded generation by 9 PWh when assuming a 20% co-firing ratio. With the installation of CCS and 20% co-firing, stranded generation could be reduced by 42–62 PWh, depending on whether half or all CCS suitable plants would be converted to CCS. The contribution of biomass co-firing is relatively small, at between 9 and 10 PWh, since the installation of CCS alone already allows for a reduction in stranded generation by 33–52 PWh (as per Fig. 3b). In Fig. 3d, with a 50% co-firing ratio and no CCS conversion, stranded generation could be reduced by 13 PWh. With a 50% co-firing ratio and CCS conversions, stranded generation could be reduced by 47–68 PWh in total, of which 14–16 PWh would come from biomass co-firing.

Therefore, we find that coal to gas conversions may reduce stranded generation by 10–30 PWh, and CCS and biomass together by 33–68 PWh. However, the estimates for coal to gas (Fig. 3a) have been calculated separately to the other conversion options (Fig. 3b–d) and cannot be directly added (see the Methods section for more details). Furthermore, there are differences in impact across models that arise from model settings on energy conversion technologies, energy technology choices, substitutability and deployment (see Supplementary Table 1 for more details).

**Impact of energy demand, alternative electricity sources, CCS and bioenergy availability.** Estimates in Figs. 2, 3 suggest that fast low-carbon technology deployment, enhanced by plant conversions, could mitigate the risk of stranding. Yet, the previous estimates have assumed that four technologies (CCS, bioenergy, solar and wind and nuclear energy) will be available and fully deployed.

Several scenarios within and outside the AMPERE database make less optimistic assumptions about technology take-up. For example, the IEA's Sustainable Development Scenario[36] assumes a late take-up of biomass with CCS (see Supplementary Fig. 6). Between 2021 to 2050 (when most asset stranding would happen for existing and in-the-pipeline plants in our database), we can calculate that the IEA scenario corresponds to total energy production from biomass with CCS of only 4 PWh. In all technology deployed scenarios of AMPERE, this technology generates an average across IAMs of 40 PWh between 2021 and 2050.

Below, we discuss the possible impact of low or late technology diffusion on the amount of stranded generation. In Fig. 4, we make different assumptions regarding technology availability. We compare the amount of stranded generation in all technologies deployed scenarios with the amount of stranded generation in scenarios that assume one of the following four technologies is unavailable or insufficiently developed: (1) CCS; (2) bioenergy; (3) solar and wind and (4) nuclear energy. We furthermore look at a scenario in which energy intensity reduces faster, at 1.5 times of the historical rates. Energy demand is, therefore, lower,

especially at the end of the century. The model-specific estimates (scatter points in Fig. 4) are directly comparable to those presented in Fig. 2. However, the average value across IAMs (the bars in Fig. 4) are not comparable because not all IAMs have run the scenarios necessary to make technology-specific pairwise comparisons. The impact of the availability of CCS, for example, can only be computed for GCAM, REMIND and MESSAGE.

Figure 4 shows that the unavailability of CCS and bioenergy (mostly combined with CCS) has a significant impact on the amount of stranded generation, increasing by 69% and by 45%, respectively on average. This is because CCS prevents emissions from going to the atmosphere, while bioenergy coupled to CCS provides a mechanism for negative emissions.

In contrast, the other low-carbon technologies studied (wind, solar and nuclear power), as well as a further reduction in energy intensity, have ambiguous impacts. Some IAMs suggest that they would reduce the amount of stranded generation. This is possible because they reduce the carbon intensity of the electricity supply, allowing for more electricity to be generated from fossil fuels. Other IAMs suggest that the development of these technologies would in fact increase the amount of stranded generation. This could be the case as wind, solar and nuclear energy compete with fossil fuels for meeting electricity demand. Greater reduction in energy intensity could reduce electricity demand and therefore lower the need for fossil fuel power plants.

A corollary finding to Fig. 4 is that no IAM implies low amounts of stranded generation unless they assume the strong deployment of CCS or bioenergy. In the absence of CCS, GCAM, REMIND and MESSAGE—three models with relatively low amounts of stranded generation in Fig. 2—would record much higher amounts of stranded generation (by 200 PWh for GCAM and 100 PWh for MESSAGE and REMIND). The amounts of stranded generation obtained with these models are also impacted by limiting bioenergy. Especially, GCAM assumes very large amounts of bioenergy combined with CCS in all technologies deployed scenarios. If bioenergy is not fully deployed, the amount of stranded generation in GCAM increases by 230 PWh.

## Discussion

Overall, our results underline a clear stranding risk for investors, plant operators and policymakers. Even if CCS and bioenergy are deployed quickly and extensively in the 21st century, an average 267 PWh of electricity generation could be at risk of stranding under a 2 °C target. If some operating and in-the-pipeline coal power plants were converted to gas, stranding could be reduced by up to 30 PWh. The adoption of CCS and the co-firing of biomass may reduce stranding by 33–68 PWh, depending on the share of CCS-suitable plants finally converted, and the assumed co-firing ratio.

Our analysis comes with limitations. Although we have collected the best available data, our unit-level power plant dataset covers 88% of the world's installed capacity in 2018. We are likely to underestimate electricity production, and therefore the amount of stranded generation. Furthermore, there is missing data on key variables like the online year and retirement year of power plants. Additionally, our analysis assumes that sufficient plants can be converted to meet an emissions budget. In reality, there may be technical and economic barriers at the individual plant level which mean our estimate of the potential to reduce stranded generation may be overestimated. Also, even though we provide lower- and upper-bound estimates for the reduction in stranded generation from plant conversions, there is much uncertainty regarding the potential and pace of future conversions.

Furthermore, our estimates of stranded generation rely on IAMs, which have long been criticised for the way they handle

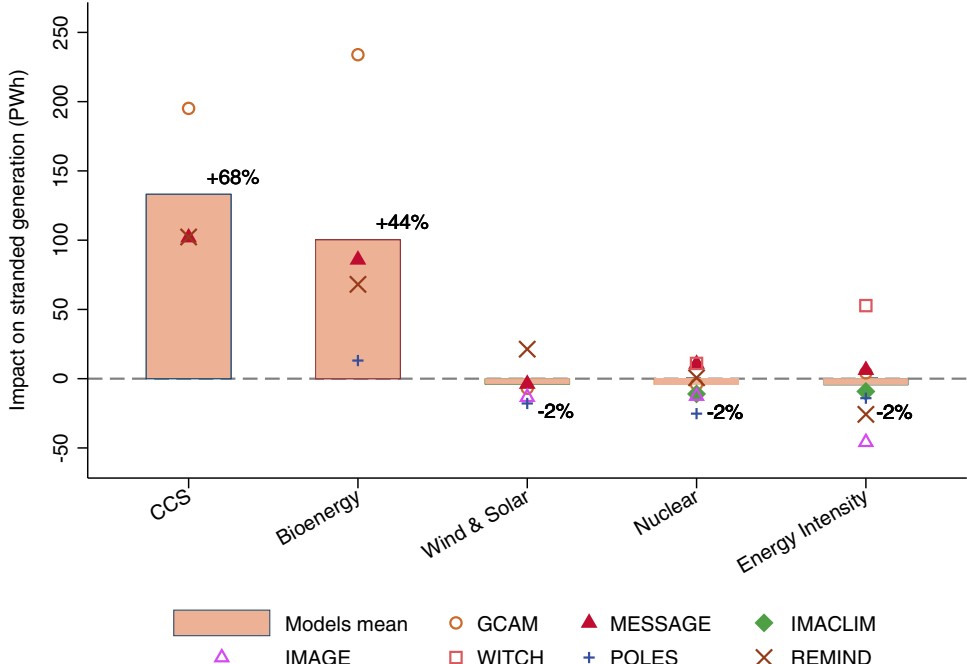

**Fig. 4 Impact of technology availabilities on global stranded generation.** Impacts are computed from the difference in stranded generation between all technologies deployed scenarios and the scenarios with one technology insufficiently deployed. Positive effects suggest that the amount of stranded generation increases when a particular technology is unavailable or insufficiently deployed. For energy intensity, the impact represented the difference between the scenarios where the energy intensity reduces at historical rates and 1.5 times historical rates. Scatter points represent the estimation of individual models and bars show the average across all integrated assessment models. The percentage values next to each bar represent the average change in stranded generation in scenarios with one technology insufficiently deployed compared to the scenarios with all technologies deployed.

uncertainty, relative prices, behaviour and other factors[21,37–40]. A recurrent problem in the literature using IAMs is that models tend to provide divergent results due to their different model settings[20,41,42]. In our analysis, the results vary significantly across IAMs. However, our analysis reconciles the findings from different models. These results provide insight as to the direction and the order of magnitude of the impact of CCS and bioenergy on stranded generation.

The amount of electricity generation at risk of stranding would rise by 69% and 45%, respectively if either CCS or bioenergy are not deployed to their full potential. These technologies with the greatest potential to reduce stranding are the ones that face the strongest challenges to their large-scale deployment. Despite some pilot and demonstration projects[43], the deployment of CCS is still hampered by the high cost of early demonstration projects and the availability of storage sites[44,45]. Similarly, the deployment of bioenergy is experiencing challenges in feedstock availability and economical cost[46], and facing disputes over sustainability, because of risks of deforestation, concerns over food security and potential biodiversity loss[22,47]. This suggests that asset exposure to stranding is likely to be much higher than estimated in all technologies deployed scenarios.

The most immediate option to reduce the risk of assets being stranded and investments being lost is to fully account for stranding risks now. Because decisions have to be made today, it may be unwise to rely on optimistic forecasts of technology deployment in the future. Most of the risk of stranding assessed in this paper relates to plants that are currently in the pipeline. If companies continue to invest in fossil fuel-based infrastructure, some of these assets risk stranding even before they are built. This puts stakeholders and policymakers in a situation in which precaution would suggest that very little or no new fossil fuel power plants can be commissioned, and existing plants may have to retire early or reduce their utilisation rate substantially.

## Methods

**Data.** Our analysis uses two categories of data: (1) global power plant data; (2) climate scenario data.

Power plant data: We use a global generator-level power plant dataset to estimate the future electricity generation from power plants that are currently operating and in the pipeline (i.e. under construction and planned). We compile generator-level data by merging the following databases: (a) Global Coal Plant Tracker (January 2019)[48]; (b) S&P Global Platt's World Electric Power Plants (WEPP) database (Q4 2019)[49]; (c) WRI's global database of power plants (June 2019)[50]. Merging several sources gives us better coverage of power plants than using only one database. For example, WEPP is known to have imperfect coverage of microgeneration and Chinese power generators, which is compensated by the data in the Global Coal Plant Tracker and WRI's database. An overview of the generator-level power plant data can be found in Supplementary Table 3. The capacity of currently operating generators in our dataset aggregates to 6349 GW, covering 88% of the global electricity capacity in the year 2018[51].

The sources are merged manually by confirming the power generator name, location, current status, online year and capacity. Online year refers to the year in which the power generator started operating. We supplemented this effort with internet searches wherever necessary. Where matched generators have conflicting fields (for example different statuses) the most recent data are used.

Climate scenario data: We use climate scenarios from the AMPERE database to determine the evolution of electricity generation throughout the end of the century. AMPERE is a modelling comparison project that integrates outputs from different IAMs to improve understanding of possible scenarios toward climate targets. AMPERE stands for Assessment of Climate Change Mitigation Pathways and Evaluation of the Robustness of Mitigation Cost Estimates.

Particularly, we use the AMPERE Working Package 2 (WP2) database, which provides pathways of electricity generation given different technology availabilities[20]. AMPERE WP2 compiles a set of around 400 scenarios to 2100, built upon 7 IAMs (GCAM, IMACLIM, IMAGE, MESSAGE-MACRO, POLES, REMIND and WITCH) that have been published in peer-reviewed journals. The AMPERE WP2 database also includes scenarios modelled by DNE21+ and MERGE-ETL models. However, these two models have been excluded in our analysis as DNE21+ only models the period through to 2050 and MERGE-ETL only models scenarios of OECD.

Supplementary Table 1 presents the general characteristics (panel a), socioeconomic drivers (panel b) and key energy technology assumptions (panel c) of these IAMs. The information is compiled from IPCC[52] and Kriegler et al.[41]. As a modelling comparison project, AMPERE harmonised GDP and population assumptions across models to facilitate the analysis of model differences. Nevertheless, models differ in numerous ways and lead to varying results, as shown

in our results. Additional details on the models used in the AMPERE project are provided in Kriegler et al.[41].

The AMPERE WP2 database models the availability of four low-carbon technology options: CCS, solar and wind, bioenergy and nuclear energy. It furthermore assumes that energy intensity reduces at either historical rates or 1.5 times historical rates. The details of the technology dimensions are presented in Table 1. The baseline scenario is an all technologies deployed scenario that assumes that all four technologies are fully available and energy intensity reduces at historical rates. Supplementary Fig. 2 displays the global electricity generation by technology in all technologies deployed scenarios of each IAM. Alongside, the AMPERE database provides other scenarios that assume each one of the four technology options is either unavailable or of limited availability. Another scenario assumes that all four technologies are available and energy intensity reduces faster, by 1.5 times historical rates. Comparisons between the all technologies deployed scenario and these scenarios allow us to estimate the impact of each technology availability on stranded generation.

Scenarios in AMPERE WP2 are constructed for different levels of climate policy stringency. We use the 450 Optimal scenarios, which require greenhouse gas levels to stabilise at 450ppm $CO_2$-equivalent. This level of stabilisation corresponds to GHG emission scenarios in the literature widely consistent with keeping long-term temperature rise below 2 °C compared to pre-industrial levels[19,20]. The 450 ppm scenarios in the AMPERE WP2 database correspond to a cumulative $CO_2$ emission budget of 1500Gt $CO_2$ from 2000 to 2100. Optimal indicates that global emissions follow an optimal pathway in which required policies are introduced immediately to meet the long-term emissions budget and no explicit short-term target for 2030 is assumed[20].

AMPERE WP2 scenarios include data at regional and country levels. The world is divided into five regions: Asia, Latin America (LAM), Middle East and Africa (MAF), OECD countries (OECD) and the Reforming Economies of the former Soviet Union (REF).

The regional data make it possible to estimate stranded generation at the regional level and then aggregate to the global level, which provides a more accurate global estimation as this does not make assumptions about exchanges of electricity between regions. If we calculated stranded generation at the global level, this may assume inter-regional exchanges of energy (e.g. between Europe and Asia) and this may mask regional stranding. The global level estimated for stranding might be lower than the sum of all regions. For example, while electricity generation from coal in OECD goes down, it may go up in Asia. This amount of reduced electricity generation in the OECD would not be stranded if it powered the Asian market, as assumed with aggregated values. On regional or country-level analysis, however, this amount would become stranded. Therefore, we calculate stranded generation at the regional level first and then aggregate them at a global level when using climate scenarios from the AMPERE database.

AMPERE WP2 also includes country-level scenarios (or clusters of countries): China, Brazil, India, USA, Japan, Russia and EU. We use these data to produce complementary country-level analyses.

**Methodological approach.** We proceed in four steps, combining a data-driven method and IAM results. Firstly, we estimate the future electricity generation from currently operating and in-the-pipeline plants (results shown in Fig. 1). Secondly, we compute stranded generation for each scenario as the difference between the future electricity generation from operating and in-the-pipeline plants, and the electricity generation in the AMPERE 450 ppm scenarios (results shown in Fig. 2). Thirdly, we recalculate stranded generation after accounting for the fact that operating and in-the-pipeline plants could be converted into less polluting assets, for example by converting a coal-fired power plant into a gas-fired power plant (results shown in Fig. 3). Finally, we compare stranded generation across technology scenarios to evaluate the impact of technology availability on stranded generation (results shown in Fig. 4).

The advantage of our method is that it allows us to understand stranded generation by using results from different IAMs. IAMs are constructed based on different structures and assumptions, therefore results from different IAMs usually differ substantially and are not directly comparable[20,41]. As we have several models corresponding to each scenario, we have several estimates for assessing stranded generation in each scenario. To address this, we present the mean value across different IAMs as well as individual model results in our figures.

Step 1: Estimating future electricity generation from operating and in-the-pipeline plants. We use our power plant data and follow the approach developed by Pfeiffer et al.[15] to estimate how much future electricity demand can be met from current plants.

We estimate yearly electricity generation from currently operating plants and those in the pipeline (planned and under construction) at the generator level and then aggregate it to the country level, the regional level and globally. We compute the electricity generation from each generator $i$ in year $t$ as follows:

$$\text{Generation}_{it} = \text{Capacity}_i \times 24 \times 365 \times \text{Utilisation}_{it} \quad (1)$$

Where $\text{Generation}_{it}$ is the annual electricity generation of generator $i$ in year $t$; $\text{Capacity}_i$ is the maximum hourly capacity of generator $i$ (which we multiply by 24 h times 365 days), and is the utilisation rate of generator $i$ in year $t$. We do not

know the exact starting date or retirement date of a generator within the year. Therefore, in the online years and retirement years, we simply assume that generators are operating for six months (the utilisation rates take half values compared with other years).

(1) *Utilisation rates.* In the absence of any climate constraint, we assume that currently operating plants and those in the pipeline would follow the utilisation rates assumed in IEA scenarios. We extract global fuel-specific load factors (i.e. utilisation rate) for power generators from IEA's World Energy Outlook 2019[51] and IEA World Energy Outlook in 2005–2019 for historical data and present them in Supplementary Fig. 7. The IEA Current Policies Scenario presents what happens if business continues as usual, without any additional policy changes, while the Stated Policies Scenario considers today's policy intentions and targets. No significant differences are found between future utilisation rates in these scenarios. We see a clear decreasing trend in historical utilisation rates of coal- and oil-fired power plants, while gas-fired power plants saw a slight increase from 2004 to 2018. Therefore, for our main results, we have assigned the average value of utilisation rates in Stated Policies Scenarios to each individual generator based on its fuel type (as presented in Column (1) in Supplementary Table 2). The assigned utilisation rates for coal-, gas-, oil- and biomass-fired power generators in the baseline are 56%, 39%, 23 and 55%, respectively. In Supplementary Fig. 4, we furthermore show the sensitivity of our results to the choice of utilisation rates by using the maximum and minimum historical utilisation rates (as presented in Columns (2) and (3) in Supplementary Table 2).

(2) *Lifetime of generators.* Since we are interested in total future electricity generation until the end of the century, we need to make assumptions about when currently operating and in-the-pipeline generators will retire. Only about 5% of power generators in our dataset contain information on when the generator is expected to be retired. For the other ones, we simulate their lifetime based on the retirement year of similar generators. We consider that generators are of the same type if they use the same fuel, unit technology type and steam type (usually either subcritical, supercritical, and ultra-supercritical steam conditions), are in the same capacity range (under 400 MW, from 400 to 500 MW, from 500 to 700 MW, from 700 to 900 MW and beyond 900 MW), and started to operate in the same year. We use the most disaggregated levels of technology types available. For example, coal-fired power plants include coal, coke, syngas from gasified coal, synthetic gas from petroleum coke, coke-oven gas, coal steam gas, coal-water slurry, corex process offgas, etc. Then, we assume that the lifetime of power generators of a given type follow a Poisson distribution with a mean value equal to the average known lifetime of generators of the same type.

For the generators that are already older than their estimated lifetime but are still operating, we assume that these power generators will progressively retire within the next 10 years. We assign them a retirement date depending on their age. To do so, we rank these generators based on their age, from the oldest (first) to the youngest (last). We then multiply their ranking, e.g. 15th oldest out of 42 (15/42), by 10 years to assign them with a remaining lifetime (in this case 15/42 times 10 is equal to 3.57 years).

We compare our assumptions of the lifetime of generators with the assumptions used previously in the literature in Supplementary Table 4. Column (1) shows the average lifetimes assumed in our baseline: 40 years for coal-fired power generators and 39 years for gas- and oil-fired power generators. Columns (2)–(4) list the lifetime assumed in key literature.

With these baseline assumptions for a lifetime, we provide estimates for the global yearly electricity generation levels in Fig. 1. We further show the sensitivity of our estimates to the assumptions of a lifetime in Supplementary Fig. 4. In Supplementary Fig. 4, the extended case assumes all fossil fuel power generators extend their lifetime by 10% compared to what we assumed in our baseline estimations while, in the reduced case, the expected lifetime is reduced by 10%.

Step 2: Estimating stranded generation for each climate scenario. We define a stranded generation as the difference between the future electricity generation from currently operating and in-the-pipeline plants and the electricity generation levels as given in the different scenarios of the AMPERE database. Therefore, our estimates of stranded generation are scenario-specific and depend on the assumed climate target (450 ppm) as well as the technologies available in the future to produce electricity. Furthermore, because the AMPERE database covers the results of different IAMs for each specific scenario, we have as many estimates as we have IAMs.

For each AMPERE scenario, we estimate the amount of stranded generation by fossil fuel types (coal, gas and oil) in each year separately and then sum up the total fossil fuel stranded generation over the period 2021–2100. The AMPERE database provides us with the yearly amounts of electricity production from fossil fuels with and without CCS in each scenario. In our estimates in Fig. 2, we assume no conversion of any operating or in-the-pipeline fossil fuel power plants to CCS. Therefore, we use the electricity production from fossil fuels without CCS in the IAM scenarios to compute the amount of stranded generation.

If the electricity generation estimated from operating and in-the-pipeline plants is larger than the electricity generation assumed in the climate scenario, then the difference corresponds to the amount of stranded generation for this year; otherwise, there is no stranded generation for this year in this scenario. We consider electricity generation from currently operating plants to supply the energy demand first, as these plants are already built.

Precisely, we estimate the stranded generation of fuel $r$ for region/country $c$ in year $t$ as follows:

$$SA_{rct,operating} = \max\{(Generation_{rct,operating} - Scenario_{rct,woCCS}), 0\} \quad (2)$$

$$SA_{rct,pipeline} = Generation_{rct,pipeline} \text{ if } SA_{rct,operating} > 0 \quad (3)$$

$$SA_{rct,pipeline} = \max\Big\{Generation_{rct,pipeline} + Generation_{rct,operating} \\ - Scenario_{rct,woCCS}), 0\Big\} \text{ if } SA_{rct,operating} = 0 \quad (4)$$

$$SA_{rct} = SA_{rct,operating} + SA_{rct,pipeline} \quad (5)$$

Where $SA_{rct,operating}$, $SA_{rct,pipeline}$ and $SA_{rct}$ are the estimated stranded generation of operating, in-the-pipeline and total plants of fuel $r$ for region/country $c$ in year $t$, respectively. $Generation_{rct,operating}$ and $Generation_{rct,pipeline}$ are the electricity generation of operating and pipeline plants of fuel $r$ for region/country $c$ in year $t$, respectively; $Scenario_{rct,woCCS}$ is the predicted level of electricity production of fuel $r$ without CCS for region/country $c$ in year $t$ in AMPERE climate scenarios.

We define stranding in terms of generation since it captures the overall impact that climate constraints could have on the amount of electricity generation that will have to be cancelled.

Step 3: Re-estimating stranded generation while taking potential plants conversion into account. In the above calculations, we have assumed that operating and in-the-pipeline power plants would not be converted to cleaner options as a way to reduce stranded generation. We relax this assumption below and re-estimate stranded generation in all technologies deployed scenarios under the assumption that some plants could be converted (the results shown in Fig. 3). We consider three conversion options: (1) coal to gas; (2) adoption of CCS in existing and in-the-pipeline plants and (3) coal to biomass.

(1) *Coal to gas conversions*. Firstly, we consider that conversions are only viable if there is a need for gas-fired power generation that cannot already be covered by existing and in-the-pipeline power plants. This is the case if the electricity from gas-fired power plants in the IAM is superior to the electricity that can be produced by existing and in-the-pipeline plants. In this case, conversions can be used to produce the remaining gas-fired power generation. The difference between the total electricity generated in the IAM and the electricity that can already be provided by existing and in-the-pipeline plants constitutes an upper bound for the conversion potential. This is given by Eq. (6):

$$ExtraGas_{ct} = \max\{(Scenario_{gas,ct,woCCS} - Generation_{gas,ct}), 0\} \quad (6)$$

$ExtraGas_{ct}$ in region/country $c$ and year $t$ is the additional power generation from gas that cannot be already satisfied with existing and in-the-pipeline gas-fired power plants. $Scenario_{gas,ct,woCCS}$ is the electricity generation from gas-fired power plants without CCS, and $Generation_{gas,ct}$ is the electricity generated from existing and in-the-pipeline gas-fired power plants.

Secondly, we consider that the technical feasibility of plant conversions might limit the quantity of electricity generated by converted plants from coal to gas. We consider that coal-to-gas conversions can only take place in countries that also have operating gas-fired power plants since the converted plants must be able to have access to gas infrastructure and supplies. In these countries, we consider that only a fraction of coal-fired power plants would be converted. We calibrate this share based on evidence from EIA[30] that 5% of coal-fired power capacity have already been converted to gas in the US. We provide results in which we assume that either up to 5 or 20% of coal-fired power capacity would be converted to gas (only in countries where there already are gas-fired power plants).

The electricity generated from converted plants in each year is then assumed to respect both boundaries: the one set by the IAM on gas demand, and the one that we set on technical feasibility. This is given by Eq. (7)

$$Conversion_{gas,ct} = \min\{Generation_{ct,coal-to-gassuitable} \times Percentage, ExtraGas_{ct}\} \quad (7)$$

$Conversion_{gas,ct}$ is the amount of electricity generated from coal-to-gas converted plants. Percentage is the conversion percentage of coal-to-gas suitable units. It is equal to 5 or 20% in our calculations.

Based on Eq. (7), we re-estimate the amount of stranded generation for coal plants by subtracting the generation that comes from converted coal-fired power plants

$$SA_{coal,ct} = \max\{(Generation_{coal,ct} - Conversion_{gas,ct} - Scenario_{coal,ct,woCCS}), 0\} \quad (8)$$

$SA_{coal,ct}$ is the stranded generation of coal-fired power plants. $Generation_{coal,ct}$ is the electricity generated from existing and in-the-pipeline coal-fired power plants, and $Scenario_{coal,ct,woCCS}$ is the electricity generated in the IAM from coal-fired power plants without CCS.

(2) Adoption of CCS in existing plants and those in the pipeline. The AMPERE scenarios distinguish between the electricity produced by plants without and with

CCS. In Fig. 2, we assumed that none of the electricity produced with CCS was coming from operating and in-the-pipeline plants. We relax this assumption.

The total energy generated from fossil fuels with CCS in the IAM scenarios sets an upper bound for CCS conversions ($Scenario_{rct,wCCS}$). Furthermore, we also look at the potential for conversions among existing plants and those in the pipeline. We use the criteria of Caldecott et al.[33] units are deemed CCS-suitable if they have a capacity above 100 MW, are less than 20 years old, emit less than $1000\,g\,CO_2/kWh$ and are located within 40 km of geological areas suitable for CCS. Using these criteria, we find that around 24% of operating and in-the-pipeline fossil fuel power units would be suitable for CCS conversion. We then assume that either half or all CCS-suitable plants could be converted to CCS.

The total share of energy produced from existing or in-the-pipeline coal-fired power plants equipped with CCS is then set to be either equal to the maximum limit set by the IAM on the energy that could be generated from CCS; or equal to the total potential from existing or in-the-pipeline plants, which we assume to be either the 24% of CCS-suitable plants, or half of these.

The corresponding equations for fuel $r$, region/country $c$ and year $t$ are

$$Conversion_{rct,CCS} = \min\{Generation_{rct,CCSsuitable} \times CCSshare, Scenario_{rct,wCCS}\} \quad (9)$$

$Conversion_{rct,CCS}$ is the amount of fossil fuel generation from plants converted to CCS from fuel $r$, in region/country $c$ in year $t$. $Generation_{rct,CCSsuitable}$ is the electricity generated from fossil fuel power plants that are CCS-suitable. CCSshare is the share of these plants that are converted to CCS. It is assumed to be equal to 50 or 100% in our calculations. Then, the amount of stranded generation with CCS conversions is equal to

$$SA_{rct} = \max\{(Generation_{rct} - Scenario_{rct,woCCS} - Conversion_{rct,CCS}), 0\} \quad (10)$$

Where $SA_{rct}$ is the estimated amount of stranded generation of fuel $r$, in the region/country $c$ in year $t$. $Generation_{rct}$ is the electricity that can be generated by all existing and in-the-pipeline power plants (for fuel $r$). $Scenario_{rct,woCCS}$ is the electricity generated from fuel $r$ without CCS in the IAM scenario.

(3) *Coal-to-biomass conversions*. The co-firing of biomass in coal power plants could reduce the amount of stranded generation. It is likely to happen in coal-fired power plants that have CCS installed. We recalculate the total amount of CCS conversions while allowing for biomass to be co-fired in coal-fired CCS-suitable plants. We, therefore, modify Eq. (9) to include biomass as a potential fuel for CCS-suitable plants. The change does not affect CCS conversions for gas- and oil-fired power plants but it has an impact on CCS conversions for coal-fired power plants. With co-firing, the electricity generated from coal-fired CCS-converted plants has to be inferior to the electricity generated from the sum of coal and biomass with CCS in the IAMs

$$Conversion_{coal,ct,CCS} \leq Scenario_{coal,ct,wCCS} + Scenario_{biomass,ct,wCCS} \quad (11)$$

Furthermore, the co-firing ratio between coal and biomass has to be respected

$$Conversion_{coal,ct,CCS} \leq Scenario_{coal,ct,wCCS}/(1 - Ratio) \quad (12)$$

Ratio is the share of biomass used to co-fire with coal. We set this ratio to be either 20 or 50%. This is because IEA and IRENA[35] estimate that a 20% co-firing ratio is feasible in most cases, while a 50% co-firing ratio is technically achievable.

The two constraints above ensure that the amount of biomass co-fired in coal-fired power plants retrofitted with CCS remain below $Scenario_{biomass,ct,wCCS}$ and below the maximum amount possible for a given co-firing ratio, i.e. $Scenario_{coal,ct,wCCS} \times Ratio / (1 - Ratio)$.

We rewrite Eq. (9) with these new constraints for the amount of CCS conversions from coal-fired power plants

$$Conversion_{coal,ct,CCS} = \min\{Generation_{coal,ct,CCSsuitable} \times CCSshare, (Scenario_{coal,ct,wCCS} \\ + Scenario_{biomass,ct,wCCS}), Scenario_{coal,ct,wCCS}/(1 - Ratio)\} \quad (13)$$

The difference between Eq. (9) and Eq. (13) corresponds to the amount of biomass co-fired in converted plants with CCS.

Furthermore, coal plants without CCS can also cofire biomass. The maximum amount of biomass co-fired in plants without CCS is equal to the electricity generation from coal without CCS ($Generation_{coal,ct}$) times the co-firing ratio. However, because some plants have been converted to CCS, we need to subtract $Conversion_{coal,ct,CCS}$ from $Generation_{coal,ct}$

$$MaxGen_{ct,cofired,woCCS} = (Generation_{coal,ct} - Conversion_{coal,ct,CCS}) \times Ratio \quad (14)$$

$MaxGen_{ct,cofired,woCCS}$ is the maximum amount of electricity from biomass without CCS that can be generated from coal-fired power plants that have not adopted CCS, given the co-firing ratio.

In addition, coal-to-biomass conversions without CCS are only viable if there is a need for power generation from biomass without CCS that is not already covered by existing and in-the-pipeline biomass power plants. This is the case if the electricity generation from biomass power plants without CCS in the IAM is superior to the electricity that can be produced by existing and in-the-pipeline

biomass power plants. This is given by Eq. (15)

$$\text{ExtraBio}_{ct,woCCS} = \max\{(\text{Scenario}_{biomass,ct,woCCS} - \text{Generation}_{biomass,ct,woCCS}), 0\}$$

(15)

$\text{ExtraBio}_{ct,woCCS}$ is the power generation from biomass without CCS that is not already satisfied with existing and in-the-pipeline biomass power plants. $\text{Scenario}_{biomass,ct,woCCS}$ is the electricity generated from biomass without CCS in the IAM scenario. $\text{Generation}_{biomass,ct,woCCS}$ is the electricity that can be generated from existing and in-the-pipeline biomass power plants. Biomass-fired power plants include plants that use waste, biogas, biomass and bio-oil as fuel. We apply the utilisation rates in Stated Policies Scenarios from IEA World Energy Outlook 2019 to biomass power generators, which is 55%, to calculate $\text{Generation}_{biomass,ct,woCCS}$.

The electricity generated from coal-to-biomass converted plants without CCS is either equal to $\text{ExtraBio}_{ct,woCCS}$ or to $\text{MaxGen}_{ct,cofired,woCCS}$

$$\text{Conversion}_{biomass,ct,woCCS} = \min\{\text{ExtraBio}_{ct,woCCS}, \text{MaxGen}_{ct,cofired,woCCS}\} \quad (16)$$

We furthermore recalculate the amount of stranded generation for coal-fired power plants

$$SA_{coal,ct} = \max\{(\text{Generation}_{coal,ct} - \text{Scenario}_{coal,ct,woCCS}$$
$$- \text{Conversion}_{coal,ct,CCS} - \text{Conversion}_{biomass,ct,woCCS}), 0\}$$

(17)

In Eq. (17), we use Eq. (13) and not Eq. (9) to compute $\text{Conversion}_{coal,ct,CCS}$.

Step 4: Estimating the impact of technology availability. We construct pairwise scenario comparisons between a) the scenarios where the four technologies are fully available; and b) the scenarios where one single technology is not available, or its diffusion is limited (as presented in Table 1). Specifically, the matched pairwise scenarios are using the same IAM (e.g. GCAM) but only have one technology assumption different (e.g. scenarios, where CCS is fully available, are paired with scenarios where CCS is not allowed, while other technology dimensions on nuclear energy, wind and solar and bioenergy potential are the same). Then we compute the difference of stranded generation (estimated in Eqs. (2)–(5)) between matched pairwise scenarios and we define this as the impact of a particular technology on stranded generation.

The AMPERE database also includes scenarios with all four low carbon technologies available, but higher reduction rates (by 50%) in energy intensity. We compare the scenarios which assume energy intensity reduction at historical rates with these higher reduction rates scenarios to calculate the impact that faster energy intensity reduction could have on the amount of stranded generation.

## Data availability
Power plant data are compiled from (a) Global Coal Plant Tracker, publicly available at: https://endcoal.org/global-coal-plant-tracker/; (b) World Electric Power Plants (WEPP) database, purchased from S&P Global Market Intelligence; (c) WRI's global database of power plants, publicly available at: https://datasets.wri.org/dataset/globalpowerplantdatabase. As part of power plant data (WEPP) are license protected, we are not able to share the raw data on power plants. The AMPERE climate scenarios data are publicly available at: https://tntcat.iiasa.ac.at/AMPEREDB/dsd?Action=htmlpage&page=about. The IPCC SR1.5 scenarios data are publicly available at: https://data.ene.iiasa.ac.at/iamc-1.5c-explorer/. The IEA Sustainable Development Scenarios regarding CCS are publicly available at: https://www.iea.org/reports/ccus-in-clean-energy-transitions/ccus-in-the-transition-to-net-zero-emissions. The main results figure source data are provided in the Figure Source Data excel file.

## Code availability
The analysis code used to produce the main results for the paper is available at https://doi.org/10.5281/zenodo.5589287.

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

## Acknowledgements

For helpful comments, the authors thank Alex Clark, Simon Dietz, Niall Farrell, Aoife Haney, Cameron Hepburn, Lucas Kruitwagen, Francois Lafond, and Xiaoyan Zhou, although this does not imply their approval or agreement with anything in this paper. The authors also thank seminar participants at INET Oxford, Queen's University Belfast, University College London, GAUC Forum, SEEDS, Post-Carbon Transition, International Academic Symposium, and IAEE conference. The authors thank the Nature Conservancy (Y.L. and F.C.), Oxford Martin School (Y.L.), China Scholarship Council (Y.L.), University of Oxford (Y.L.), China Oxford Scholarship Fund (Y.L.), Oxford Net Zero (S.S.), University of Oxford's Strategic Research Fund (S.S.) for financial support.

## Author contributions

Y.L. contributed to the initial conception and design of the work, data collection, analysis and interpretation, drafting the article and critical revision of the article. F.C. contributed to the initial conception and design of the work, data analysis and interpretation, drafting the article and critical revision of the article. S.S. contributed to data analysis and interpretation, drafting the article and critical revision of the article. A.P. contributed to the initial conception and design of the work, data collection, analysis and interpretation and critical revision of the article.

## Competing interests

The authors declare no competing interests.
