## [Peer Review File · Nature Communications]

nature portfolio

Peer Review FileREVIEWER COMMENTS

Reviewer #1 (Remarks to the Author):

Overall comments

The topic treated in paper is relevant and important. One could argue that the results are rather obvious and that one does not need models to draw the conclusions put forward in the paper. Yet, I think important to convey the conclusions drawn. It is not clear to reader how the concept of stranded assets was defined. Also, some of the assumptions on how coal power plants can be converted/replaced seem very academic and made without knowledge on power plant technology (or possibly that it is expressed in an unclear way in paper).

Detailed comments

Abstract. Last sentence reads: "Fossil fuel plants remain at substantial risk of stranding under optimistic assumptions, and even more so if CCS and bioenergy are deployed at scale."

Perhaps this should read

"Fossil fuel plants remain at substantial risk of stranding under optimistic assumptions, and even more so if CCS and bioenergy are NOT deployed at scale."?

Page 2. Not clear what the references given in the following sentence refer to.

"This paper assesses the potential for new technologies and plant conversions to reduce the need for assets to be "stranded", i.e. underused or retired early to ensure that global warming remains below 2°C (McGlade and Ekins, 2015; Caldecott et al., 2016; Carbon Tracker Initiative, 2017; Mercure et al., 2018; Pfeiffer et al., 2018)."

Is it meant that the references listed have defined what are stranded assets? Not clear, I think.

Pages 3 and 4. From the text and the figure caption in Figure 1 it is not really clear what is shown in figure. Figure caption states "Forecasted electricity generation between 2021 and 2100...". But I am not sure the word forecast is the right word. It is based on some scenario analysis, I think. Or is it meant the amount that can be produced from fossil fuels with the same fuel mix as of today's power plant mix? Not clear.

On page 4 it is stated that "Stranded assets are simply computed as the difference between the electricity generation allowed in the climate scenarios..."

But the amount of electricity that can be generated in a 2°C scenario depends on technology. I guess it is meant how much electricity can be produced from the existing fossil fuel power plant mix assuming it is operated at the same full load hours as today. Not clear, I think. Also, with an increased cost of emitting CO2 (as would be required in a scenario meeting the climate targets), the full load hours will/can go down (at least for a certain share of the power plants) as other climate neutral power generation is phased into the system.

In the text related to Figure 2 and the "all technologies deployed" scenario it becomes a bit confusing when it is stated "We assume no conversion of any existing fossil fuel fired power plants to CCS, and no conversion of coal-fired power plants to bioenergy or gas." To me application of CCS would be part of an all technology deployment scenario.

On page 6 it is stated "GCAM is the only IAM with very low levels of stranded assets. with all the other models entailing at least 190PWh of stranded assets." But with no further comment. As a reader one expects an explanation of the great difference, especially since the overall conclusions on paper rests on that there is quite many plants that must be stranded if to comply with climate targets.

The assumption given on page 6 as one case, i.e. that "First, we assume that coal-fired power plants could be converted into gas-fired power plants." seems strange. Coal plants and natural gas plants represent rather different technologies (coal condensing plants vs combined cycles) with fundamentally different boiler designs, fuel handling systems and fuel infrastructure. Not clear and assumption should be justified. Perhaps the authors mean that the coal plants are phased out and the entire plant is replaced by a gas fired plant (combined cycle) which will require access to

natural gas instead of coal. But then one wonder why not convert to renewable electricity generation or nuclear power?

Page 7/8 it is stated "These technologies may, in fact, not be available or fully deployed in the future. Furthermore, these scenarios assume a reduction in energy intensity consistent with historical improvement rates. Assuming a different level in energy intensity improvement rates could also impact the amount of stranded assets."

This sounds rather obvious; if we assume another future, the results will differ so to say.

On page 9 it is stated

"Fig. 4 shows that the availability of CCS and bioenergy (mostly combined with CCS) have a significant impact on the amount of stranded assets."

Yet, it is not clear what limits the application of CCS cost or limitations on storage sites. Clearly CCS could theoretically remove some 90% of the CO2 emissions from the fossil (and biomass) fire plants. In reality probably limited by access to storage. Although storage potential is big it is not available uniformly across the globe.

On the same page it is stated:

"In contrast, the other low carbon technologies studied (wind and solar; nuclear energy), as well as a further reduction in energy intensity, have ambiguous impacts. Some IAMs suggest that they would reduce the amount of stranded assets. This is possible because they reduce the carbon intensity of the electricity supply, allowing for more electricity to be generated from fossil fuels. Other IAMs suggest that the development of these technologies would in fact increase the amount of stranded assets. This could be the case as wind, solar and nuclear energy are competing with fossil fuels for meeting electricity demand."

To a reader it does not become clear what was the objective function in the IAMs. Surely renewable and nuclear power can replace fossil fuel power plants. To what extent the plants become stranded assets depend on their assumed economic lifetime. Should be clarified in paper how these parameters are treated.

Reviewer #2 (Remarks to the Author):

This paper looks at how converting fossil fuel power plants to low-carbon technologies may lower the risks of stranded assets to achieve global 2C goal. This is a very interesting topic that can be of great interest to both the research and policy communities. However, due to strong assumptions and simplified methodology, results of this study tend to overestimate the potential of plant conversion as an approach to lower stranded assets, and the conclusions can be misleading.

Most importantly, strong assumptions are made regarding the potential of fossil plant conversions, which is estimated at the highly aggregate level. It is highly uncertain how many existing and planned plants can be converted. For example, feasibility of CCS plants depend on carbon storage capacity, and feasibility of biomass plants depend on access to biomass resources. Decision making of plant conversion also largely depends on economic viability. How is the retrofitting cost taken into account? Does it make economic sense to retrofit existing plants to lower stranded assets? The generic conversion rate between 0% to 100% does not provide meaningful implication, and more granular analysis is needed to be able to answer the research question.

Second, the definition of asserting stranding is misleading. Stranded assets are mostly relevant to capacity, instead of generation. Lower utilization than today does not necessarily indicate assets stranding. Instead, the authors should look at premature retirement.

Third, Fig.1 shows the committed generation from existing plants are much lower than the 2C pathway, and it actually also allows new plants to be added. Showing the 2C pathway of total fossil plants may create confusion. Literature already suggests any new addition of coal plants is inconsistent with global 1.5/2c climate goals. It is important to show the pathways separately for each plant fuel type.

Fourth, the authors should look at 1.5C scenarios to be more relevant with ongoing policy discussion. Only looking at the 2c target may generate misleading messages. The authors may also consider using IAM results from more recent multi-model comparison projects. The AMPERE results seem to be a little bit outdated.

Reviewer #3 (Remarks to the Author):

Review of "Can plant conversions and abatement technologies prevent asset stranding in the power sector?"

Key results: This paper estimates the stranded assets in the fossil fuel power plant sector after adjusting for future technologies that might mitigate GHG emissions from those plants. These adjustments create the best-case scenario of for power plant utilization through 2100. The estimates show that even with full deployment of technological advances (some of which are not commercially viable at this time) about 50% of existing and future capacity in the sector will be stranded.

This is an important topic because decarbonization needs to be done in the most cost-effective way possible. Having the best possible estimates of asset stranding will help design the best decarbonization pathways. Moreover, these results are important to financiers when considering funding power plant construction. Banks increasingly face pressure to not finance new power plant construction, but evidence that shows the extent of stranded asset risk may be more persuasive than calls to join the fight against climate change.

Finally, the authors additions of plant conversions from coal-to-gas and then to BECCS help reconcile the highly different results about stranded assets from these models.

Validity: The technical analysis seems sound.

Originality and significance: This is the first paper I am aware of that carefully examines the potential effect of technology on stranded asset risk. As mentioned above, for that reason it is a significant contribution to the decarbonization conversation and should help inform policy decisions.

Data & methodology: The data and approach seem sound and the authors carefully explain the possible limitations of their data.

Appropriate use of statistics and treatment of uncertainties: NA

Conclusions: The conclusions and data interpretation are robust and valid.

Suggested improvements: There are two areas that need to be addressed.

First, the authors analysis is based on seven IAMs with four technologies "fully deployed." The IAMs give very different results about stranded assets. On page 5 in a paragraph that begins, "The amount of Stranded assets ...," the authors explain that this has to do with the structure and assumptions of the models. Later, on page 8 at the end of a paragraph that begins, "In Fig. 4, ...", the authors state that the impact of CCS can only be computed for the GCAM, REMIND and MESSAGE models. This caveat is mentioned again on page 10, in a paragraph that begins, "Furthermore, our estimates ..." It appears that across the IAMs "fully deployed" means very

different things. To provide clarity for readers it would be very helpful to create a table that shows the sorts of technologies each IAMs supports and possibly the time-frame and rate of innovation of these models. A final comment about each model regarding its approach to technological innovation, e.g., endogenous/exogenous, and how diffusion and/or rates of improvement are incorporated in the model, would help readers better understand the variability across the IAM results.

Carefully explaining what "fully deployed" means in each model pops up in Figure 3. Figure 3 shows the reduction in asset stranding when power plants are converted from coal to gas and coal to biomass. Several models show almost no reduction in stranded assets suggesting that these conversions are part of the full deployment of technology built into the models.

My second concern or preference is also related to the deployment inherent in the IAMs and thereby in the results. Can the authors somehow link their results to well-known forecasts or energy scenarios such as the IEA's assumptions about CCS or CCUS in its models. For example, in the IEA's CCUS in Clean Energy Transitions, September 2020 (<https://www.iea.org/reports/ccus-in-clean-energy-transitions/ccus-in-the-transition-to-net-zero-emissions>) a table shows cumulative CO2 capture from the power generation sector of 87,529 Mt. Is there any way to relate this assumption of the IEA SDS to the results the authors are presenting? My question is, How does the IEA assumption compare to the "fully deployed" assumptions of the IAMs used in this study?

My final comment is that the authors seem to under-sell what they have done. On page 10, in a paragraph that begins, "Furthermore, our estimates ...," there is a sentence that I didn't understand or notice the first couple of readings of the paper. It says, "Therefore, our analysis reconciles the findings from different models." If I understand Figures 2 and 3, together they show this reconciliation of results. Figure 3 adds technologies to models in which they were originally under-represented. For example, in Figure 2 the WITCH model predicts stranded assets of about 250 Pw of capacity. But in Figure 3, with plant conversions and CCS, this model shows a reduction of stranded assets of about 180 Pw at a 100% conversion ratio. I think it would be very powerful to combine these results and show the convergence of the models. Of course, a 100% conversion to BECCS is unlikely because of the lack of CCS commercialization and penetration, the lack of biomass fuels in some regions, and financial constraints, which are the "technical and economic barriers" the authors mention on page 10. The authors could then discuss asset stranding at lower conversion rates and its implications to future investment in fossil fuel fired power plants.

References: The manuscript references are thorough and appropriate.

Clarity and context: The abstract's last sentence is a bit confusing. The sentence reads, "Fossil fuel plants remain at substantial risk of stranding under optimistic assumptions, and even more so if CCS and bioenergy are deployed at scale." I am not sure what "at scale" refers to. Maybe change this to something related to "at current rates of innovation" or "estimated rates of technological penetration"?

There is a small typo on page 8 in the paragraph that begins, "Fig 4 shows ...". "The last sentences includes " ... to the atmosphere as the it stores the .., which has an extra word.

POINT-BY-POINT RESPONSE TO THE REVIEWERS' COMMENTS

We would like to take this opportunity to express our sincere thanks to the reviewers for their extremely helpful comments. The detailed responses below aim to address each comment (authors' responses in black, right after each comment in blue). We also provide a clean version and a manuscript with changes in highlight. Note that changes due to typos, footnotes and other minor rearrangements are not highlighted. We hope that all three reviewers will be satisfied with the changes.

SHORTCUTS TO REVIEWERS' COMMENTS

Reviewer #1	1
Reviewer #2	15
Reviewer #3	23

Reviewer #1

Overall comments

The topic treated in paper is relevant and important. One could argue that the results are rather obvious and that one does not need models to draw the conclusions put forward in the paper. Yet, I think important to convey the conclusions drawn. It is not clear to reader how the concept of stranded assets was defined. Also, some of the assumptions on how coal power plants can be converted/replaced seem very academic and made without knowledge on power plant technology (or possibly that it is expressed in an unclear way in paper).

Authors' response: Thank you very much for these comments. In this revised draft, we define more clearly the concept of stranded assets at the onset (on Page 2 in the manuscript):

“This paper assesses the potential for new technologies and plant conversions to reduce the need for assets to be “stranded”, i.e. underused or retired early to ensure that global warming remains below 2°C (Caldecott et al., 2016a). We follow Pfeiffer et al. (2018) and define stranded assets in the power sector in terms of stranded generation: the reduction in electricity generation from fossil fuels arising from the necessary underuse and early decommission of power plants consistent with a 2°C scenario.”

We also profoundly changed the section on plant conversions, formulating more practical hypotheses regarding the technical feasibility and potential of (1) coal-to-gas conversion; (2) CCS conversion; and (3) coal-to-biomass conversion. For your convenience, we have reproduced the new section on plant conversions below. Technical appendices and formulas to produce figure 3 have also been modified accordingly and thoroughly as described in the Methods sections (Pages 31-36).

The updated section on plant conversions read as below (Pages 7-10):

“

Impact of plant conversions

Our analysis above suggests high levels of stranded assets under all IAMs apart from GCAM. These estimates may be overly restrictive, however, because they do not consider that power plants could adapt to the risk of stranded assets by converting to use less carbon-intensive technologies. In Fig. 3 we allow for power plants to be converted and estimate the impact that this would have on stranded

assets (compared to Fig. 2). We consider three conversion options: 1) conversion from coal to gas; 2) the installation of CCS in existing plants and in those currently in the pipeline; and 3) the use of biomass in coal-fired power plants.

Coal-to-gas conversion is technically feasible and has been undertaken in more than 80 coal-fired power plants in the US between 2011 and 2019 (EIA, 2020), representing about 5 percent of total US coal-fired power capacity. In Fig 3.a, we estimate the change of stranded assets when considering potential conversions from coal to gas. We consider that coal-fired power plants located in countries that also include operating gas-fired power plants are suitable for conversion since they have access to gas infrastructure and supplies. We then assume that between 5 percent and 20 percent of them could be converted. The 5 percent corresponds to the US conversion percentage between 2011 and 2019. 20 percent is an optimistic upper bound equivalent to four times this conversion percentage (note that conversions may accelerate and could happen over a longer time frame than only 9 years). We furthermore limit possible plant conversions so that the amount of electricity produced from converted plants remains equal or below the electricity produced from gas in the IAMs. Beforehand, we have subtracted the amount already covered by existing gas-fired power plants. Results in Fig. 3.a suggest that coal-to-gas conversions may reduce stranded generation by 10 PWh to 30 PWh on average across all models.

In Fig 3.b, we look at CCS adoption from operating and in-the-pipeline fossil fuel power plants. Caldecott et al. (2016b) assesses that existing generation units are likely to be suitable to CCS installation if they have a capacity above 100MW, are less than 20 years old, emit less than 1000g CO₂/kWh and are located within 40km of geological areas suitable for CCS. Using these criteria, we find that around 24 percent of operating and in-the-pipeline fossil fuel power units would be suitable for CCS conversion. In Fig. 3.b, we show the impact of CCS conversion on stranded assets if either half of those suitable capacities, or all of them, would be equipped with CCS. We furthermore make sure that CCS installation in existing and in-the-pipeline power plants coincides with the take-up of CCS in the IAMs. Therefore, we bound our estimates of electricity generated from converted plants with CCS to be equal or inferior to the total volume of electricity generated from fossil fuels with CCS in the IAMs. The results displayed in Fig. 3.b imply that stranded generation can be reduced by 33 PWh to 52 PWh thanks to the future installation of CCS in operating and in-the-pipeline plants. Reductions are most pronounced in the IAMs in which we found the largest amounts of stranded assets (i.e. POLES, IMACLIM and WITCH).

In Fig. 3.c and 3.d, we consider that coal-fired power plants (with and without CCS) could be adapted to co-fire biomass. So far, more than 150 power plants have fired coal along with biomass, with the majority located in northern Europe and the US (Fernando, 2012). However, in the IAMs, 95 percent of the electricity generated with biomass would come from plants equipped with CCS in the future. Our calculations therefore have to consider the co-firing of biomass in some plants that are not equipped with CCS, and the co-firing of biomass in plants that would be retrofitted with CCS. Fig. 3.c. and 3.d. consider three scenarios of CCS retrofits: no plants are converted to CCS; 50 percent of CCS suitable capacities are converted to CCS; and all CCS suitable capacities are converted to CCS. The last two assumptions are the same as in Fig. 3.b. Furthermore, we allow coal plants to co-fire with biomass. Co-firing ratios can vary. The IEA and IRENA (2013) estimates that a 20 percent co-firing ratio is feasible in most cases, while a 50 percent co-firing ratio is technically achievable. Fig. 3.c. assumes a co-firing ratio of 20 percent and Fig. 3.d a co-firing ratio of 50 percent.

In both figures, the coal-to-biomass potential is restricted to the availability of sustainable biomass feedstock within each IAM. Co-firing without CCS cannot exceed the electricity generated from biomass without CCS in the IAMs, after accounting for the electricity that would already be generated from existing biomass plants. Co-firing with CCS cannot exceed the electricity generated from biomass with CCS in the IAMs. We furthermore respect the co-firing ratios. Therefore, biomass co-firing has to be equal or inferior to either 20 percent or 50 percent of the total of energy generated from converted plants: there cannot be more biomass co-fired than the amount of coal allows for.

In the absence of any CCS conversion, Fig. 3.c suggests that biomass co-firing could reduce the amount of stranded assets by 9 PWh when assuming 20 percent co-firing. With the installation of CCS and 20 percent co-firing, asset stranding could be reduced by 42 to 62 PWh, depending on whether half or all CCS suitable plants would be converted to CCS. The contribution of biomass co-firing is relatively small, at between 9 and 10 PWh, since the installation of CCS alone already allows for a reduction in stranded assets by 33 PWh to 52 PWh (as per Fig. 3.b). In Fig 3.d, with a 50 percent co-firing ratio and no CCS conversion, asset stranding could be reduced by 13 PWh. With a 50 percent co-firing ratio and CCS conversions, asset stranding could be reduced by 47 to 68 PWh in total, of which 14 to 16 PWh would come from biomass co-firing.

Therefore, we find that coal to gas conversions may reduce asset stranding by 10 to 30 PWh, and CCS and biomass together by 33 to 68 PWh. However, the estimates for coal to gas and the other conversion options (of Fig. 3.b to 3.d) have been obtained separately and cannot be directly added up (see the Methods section for more details). Furthermore, there are differences in impact across models that arise from model settings on energy conversion technologies, energy technology choices, substitutability and deployment (see Appendix Table AT2 for more details).

Fig. 3: Impact of plant conversions on global asset stranding. Scatter points represent the estimation from individual models and bars show the model mean. The conversion percentages of Figure 3.a correspond to the share of coal-fired power plants converted to gas. Those of Fig. 3.b to 3.d correspond to the share of CCS suitable plants converted to CCS. Biomass co-firing ratios vary from 0 in Fig. 3.b to 50 percent in Fig. 3.d.

We have updated references to our results on plant conversions in the discussion section (Pages 12-13):

“If some operating and in-the-pipeline coal power plants were converted to gas, stranding could be reduced by up to 30 PWh. The adoption of CCS and the co-firing of biomass may reduce stranding by 33 to 68 PWh, depending on the share of CCS-suitable plants finally converted, and the assumed co-firing ratio.”

References:

EIA (2020) More than 100 coal-fired plants have been replaced or converted to natural gas since 2011, EIA. Available at: <https://www.eia.gov/todayinenergy/detail.php?id=44636> (Accessed: 7 September 2020).

Fernando, R. (2012) Cofiring high ratios of biomass with coal, IEA Clean Coal Centre.

IEA and IRENA (2013) Biomass Co-firing in Coal Power Plants. Available at: https://iea-etsap.org/E-TechDS/PDF/E21IR_Bio-cofiring_PL_Jan2013_final_GSOK.pdf.

Caldecott, B. et al. (2016b) Stranded Assets and Thermal Coal An analysis of environment-related risk exposure. Oxford.

Detailed comments

Abstract. Last sentence reads: “Fossil fuel plants remain at substantial risk of stranding under optimistic assumptions, and even more so if CCS and bioenergy are deployed at scale.”

Perhaps this should read

“Fossil fuel plants remain at substantial risk of stranding under optimistic assumptions, and even more so if CCS and bioenergy are NOT deployed at scale.”?

Authors’ response: We thank you for spotting this typo. We amended the sentence with “*if CCS and bioenergy are not deployed at scale.*”

Page 2. Not clear what the references given in the following sentence refer to. “This paper assesses the potential for new technologies and plant conversions to reduce the need for assets to be “stranded”, i.e. underused or retired early to ensure that global warming remains below 2°C (McGlade and Ekins, 2015; Caldecott et al., 2016; Carbon Tracker Initiative, 2017; Mercure et al., 2018; Pfeiffer et al., 2018).” Is it meant that the references listed have defined what are stranded assets? Not clear, I think.

Authors’ response: We have clarified the use of these references. Precisely, Caldecott et al. (2016) defined stranded assets in a general context and Pfeiffer et al. (2018) made the definition explicit in the context of the power sector. We revised this sentence and clarified the definition of stranded assets on Page 2:

“This paper assesses the potential for new technologies and plant conversions to reduce the need for assets to be “stranded”, i.e. underused or retired early to ensure that global warming remains below 2°C (Caldecott et al., 2016a). We follow Pfeiffer et al. (2018) and define stranded assets in the power sector in terms of stranded generation: the reduction in electricity generation from fossil fuels arising from the necessary underuse and early decommission of power plants consistent with a 2°C scenario.”

The other three references correspond to prior assessments of the amount of “stranded assets”. We moved these to the third paragraph of the introduction, alongside with other references quantifying the amount of stranded assets. The revised text reads as follows on Page 2:

“Compared to prior assessments (Davis, Caldeira and Matthews, 2010; Davis and Socolow, 2014; McGlade and Ekins, 2015; Carbon Tracker Initiative, 2017; Mercure et al., 2018; Pfeiffer et al., 2018; Tong et al., 2019; Binsted et al., 2020), that have focused on calculating committed emissions from existing infrastructure and the impact of climate policy on stranding assets, this paper accounts for the possible response of the industry towards developing technological solutions when facing the risk of assets being stranded.”

Pages 3 and 4. From the text and the figure caption in Figure 1 it is not really clear what is shown in figure. Figure caption states “Forecasted electricity generation between 2021 and 2100...”. But I am

not sure the word forecast is the right word. It is based on some scenario analysis, I think. Or is it meant the amount that can be produced from fossil fuels with the same fuel mix as of today's power plant mix? Not clear.

Authors' response: We use the available information on the stock of existing and in-the-pipeline power units, as well as assumptions on asset lifetime and average utilization rates to forecast the electricity that they will generate between 2021 and 2100. This is why we used the term "forecast". In the revised version, we replaced it with the term "estimated", and we clarified the caption of Fig. 1 on Page 4:

"Fig. 1: Estimated electricity generation between 2021 and 2100, by fuel (a) and by region (b). We estimate the amount of future electricity that can be generated from currently operating power plants and those in the pipeline. Power units are assumed to be operating at the same utilisation rates as those forecasted in the IEA Stated Policy Scenarios (with a breakdown by category of power units described in Appendix Table AT4) until the end of their lifetime."

On page 4 it is stated that "Stranded assets are simply computed as the difference between the electricity generation allowed in the climate scenarios..." But the amount of electricity that can be generated in a 2C scenario depends on technology. I guess it is meant how much electricity can be produced from the existing fossil fuel power plant mix assuming it is operated at the same full load hours as today. Not clear, I think. Also, with an increased cost of emitting CO₂ (as would be required in a scenario meeting the climate targets), the full load hours will/can go down (at least for a certain share of the power plants) as other climate neutral power generation is phased into the system.

Authors' response: Thank you for the clarifying comment. You are correct: the amount of electricity generation and stranded assets is computed by fuel. We assign fuel-specific utilisation rates to power generators, namely 56%, 39% and 23% for coal, gas and oil respectively. These rates are extracted from IEA Stated Policies Scenarios, which take into account of today's policy intentions and targets.

Stranding is computed as the difference between the expected utilization of fossil fuel generation capacity under today's policy and the actual utilization to be complied with climate constraint. The actual utilization is decided by the energy mix predicted in the IAMs. These predictions take into account changes in the cost of CO₂ to reach the 2 degrees target. For example, if utilisation today is at 50 percent but could not go beyond 20 percent in the future, then we consider that 30 percent of the electricity that the plant could generate in the future would have to be stranded (50 minus 20).

Following the reviewer's suggestion, we added the following text in the caption of Figure 2 on Page 6: "*Stranded assets are computed by fuel and then aggregated.*"

In the text related to Figure 2 and the "all technologies deployed" scenario it becomes a bit confusing when it is stated "We assume no conversion of any existing fossil fuel fired power plants to CCS, and no conversion of coal-fired power plants to bioenergy or gas." To me application of CCS would be part of an all technology deployment scenario.

Authors' response: Thank you for your comment. In Fig. 2, we assume no conversion of existing power plants (i.e. currently operating or in-the-pipeline). The "all technologies deployed" scenario includes fossil-fuel power plants with CCS, and we assume that these plants will be new and different from the existing ones. To clarify, we revised the text in the caption of Fig.2 on Page 6:

"We assume no conversion of the currently operating and in-the-pipeline fossil-fuel power plants to CCS, and no conversion of the currently operating and in-the-pipeline coal-fired power plants to bioenergy or gas."

On page 6 it is stated "GCAM is the only IAM with very low levels of stranded assets. with all the other models entailing at least 190PWh of stranded assets." But with no further comment. As a reader one

expects an explanation of the great difference, especially since the overall conclusions on paper rests on that there is quite many plants that must be stranded if to comply with climate targets.

Authors' response: Thank you for your comment. We added the following text on Page 7 to explain that this is due to very high bioenergy use in GCAM scenarios:

“GCAM is the only IAM with very low levels of stranded assets, with all the other models entailing at least 180PWh of stranded assets. This is due to high levels of bioenergy deployment in GCAM scenarios: biomass with CCS (BECCS) supplied electricity could reach as much as 180 EJ per year in 2100 in GCAM, while other models only forecast a production level of 6 to 60 EJ per year from BECCS (see Appendix Fig. AF1). The biomass potential assumed in GCAM is higher than the global estimate of sustainable biomass potential of 100 EJ per year (Creutzig et al. 2015). More electricity supplied from biomass with CCS allows more electricity to be generated from fossil fuels, and thus leads to fewer stranded assets.”

The assumption given on page 6 as one case, i.e. that “First, we assume that coal-fired power plants could be converted into gas-fired power plants.” seems strange. Coal plants and natural gas plants represent rather different technologies (coal condensing plants vs combined cycles) with fundamentally different boiler designs, fuel handling systems and fuel infrastructure. Not clear and assumption should be justified. Perhaps the authors mean that the coal plants are phased out and the entire plant is replaced by a gas fired plant (combined cycle) which will require access to natural gas instead of coal. But then one wonder why not convert to renewable electricity generation or nuclear power?

Authors' response: The US EIA reported that more than 80 coal-fired plants in the US have been converted to natural gas since 2011 (EIA, 2020) and similar conversion projects have been implemented in China (Largue,2021). In the manuscript, we simply report the following (Page 7):

“Coal-to-gas conversion is technically feasible and has been undertaken in more than 80 coal-fired power plants in the US between 2011 and 2019 (EIA, 2020), representing about 5 percent of total US coal-fired power capacity.”

We do not consider switching from coal to renewable (except biomass), or coal to nuclear, as conversion options because they are fundamentally different power units. Depending on the method to convert coal to gas, the process only needs to modify and add new equipment while the majority of installations could still be used (Bedilion, 2017).

References:

EIA (2020) More than 100 coal-fired plants have been replaced or converted to natural gas since 2011, EIA. Available at: <https://www.eia.gov/todayinenergy/detail.php?id=44636> (Accessed: 7 September 2020).

Largue, P. (2021) GE completes coal-to-gas conversion at 661MW plant in China, Power Engineering International. Available at: <https://www.powerengineeringint.com/gas-oil-fired/ge-completes-coal-to-gas-conversion-at-661wm-plant-in-china/> (Accessed: 5 May 2021).

Bedilion, R. (2017) ‘Technology considerations for coal-to-natural gas conversions’, in EIA Energy Conference. Available at: https://www.eia.gov/conference/2017/pdf/presentations/robin_bedellin.pdf.

Page 7/8 it is stated “These technologies may, in fact, not be available or fully deployed in the future. Furthermore, these scenarios assume a reduction in energy intensity consistent with historical improvement rates. Assuming a different level in energy intensity improvement rates could also impact the amount of stranded assets.” This sounds rather obvious; if we assume another future, the results will differ so to say.

Authors’ response: On Page 10, we introduce Fig. 4 differently:

“Several scenarios within and outside the AMPERE database make less optimistic assumptions about technology take-up. For example, the IEA’s Sustainable Development Scenario (IEA, 2020) assumes a late take-up of biomass with CCS (see Appendix Fig. AF2). Between 2021 to 2050 (when most asset stranding would happen for existing and in-the-pipeline plants in our database), we can calculate that the IEA scenario corresponds to a total energy production from biomass with CCS of only 4 PWh. In the “all technology deployed” scenarios of AMPERE, this technology generates an average across IAMs of 40 PWh between 2021 and 2050.

Below, we discuss the possible impact of low or late technology diffusion on the amount of stranded assets. In Fig. 4, we make different assumptions regarding technology availability.”

On page 9 it is stated: “Fig. 4 shows that the availability of CCS and bioenergy (mostly combined with CCS) have a significant impact on the amount of stranded assets.” Yet, it is not clear what limits the application of CCS cost or limitations on storage sites. Clearly CCS could theoretically remove some 90% of the CO₂ emissions from the fossil (and biomass) fire plants. In reality probably limited by access to storage. Although storage potential is big it is not available uniformly across the globe.

Authors’ response: In this manuscript, the overall potential of CCS deployment is directly derived from the predictions of each IAM. Figure AF1 (added on Page 25), reproduced below, displays the global electricity generation by technology in the “all technologies deployed” scenarios of each IAM, including for CCS. These predictions account for the technical and economic difficulty of CCS deployment. To clarify the assumptions made in the IAMS, we have added a detailed summary table in the Methods chapter (Table AT2 on Pages 19-22), presenting the general characteristics, socio-economic drivers and key technology assumptions of these IAMs.

Fig. AF1: Global electricity generation from different technologies between 2021 and 2100 in AMPERE “all technology deployed” scenarios (450ppm optimal scenarios)

“Table AT2 presents the general characteristics (panel a), socioeconomic drivers (panel b) and key energy technology assumptions (panel c) of these IAMs. The information is compiled from IPCC (2014) and Kriegler et al.(2015). As a modelling comparison project, AMPERE harmonized GDP and population assumptions across models to facilitate the analysis of model differences. Nevertheless, models differ in numerous ways and lead to varying results, as showed in our results. Additional details on the models used in AMPERE project are provided in Kriegler et al.(2015).”

Table AT2: Overview of IAM models used in this study

a) General characteristics

Model name	Institution	Reference	Equilibrium type	Modelling approach	Time horizon
GCAM	JGCRI, US	JGCRI (2010)	Partial	Recursive dynamic	2100
IMACLIM	CIRED and SMASH, France	Sassi et al. (2010)	General	Recursive dynamic	2100
IMAGE	UU and PBL, Netherlands	Bouwman et al. (2006)	Partial	Recursive dynamic	2100
MESSAGE	IIASA, Austria	Messner et al. (2000)	General	Intertemporal optimization	2100
POLES	EC-JRC, Belgium	IPTS (2010)	Partial	Recursive dynamic	2100
REMIND	PIK, Germany	Leimbach et al. (2010)	General	Intertemporal optimization	2100
WITCH	FEEM and CMCC, Italy	De Cian et al. (2009)	General	Intertemporal optimization	2100

b) Social economic drivers

Model name	Exogenous drivers	Endogenous drivers
GCAM	Population, GDP, labour participation rate, labour productivity	None
IMACLIM	Labour productivity, energy technical progress, population, active population	None
IMAGE	Exogenous GDP, GDP per capita, population	Energy demand, renewable price, fossil fuel prices, carbon prices, technology progress, energy intensity, preferences, learning by doing, agricultural demand, value added
MESSAGE	Labour productivity, energy technical progress, GDP per capita, population	None
POLES	Exogenous GDP, population	Value added, mobility needs, fossil fuel prices, buildings surfaces
REMIND	Labour productivity, energy efficiency parameters of the production function, population	Investments in industrial capital stock and specific energy technology capital stocks, endogenous learning-by-doing for wind and solar power as well as electric and fuel cell vehicle technologies (global learning curve, internalized spillovers).
WITCH	Total factor productivity, labour productivity, capital technical progress	None

c) Key energy technology assumptions

Model name	Energy Conversion technologies	Energy technology choice, substitutability and deployment
GCAM	CHP, hydrogen from coal, oil, gas, and biomass, w/o and w/ CCS, nuclear and solar thermochemical, fuel to gas, coal to gas w/o CCS, biomass (w/o and w/ CCS), fuel to liquid, coal to liquids (w/o and w/ CCS), gas to liquids (w/o and w/ CCS), biomass to liquids (w/o and w/ CCS)	Logit choice model with usually high substitutability
IMACLIM	Fuel to liquid	Logit choice model, mostly high substitutability in some sectors and mostly low substitutability in other sectors, expansion and decline constraints, system integration constraints
IMAGE	CHP, hydrogen	Logit choice model, mostly high substitutability, expansion and decline constraints, system integration constraints
MESSAGE	CHP, hydrogen, fuel to gas, fuel to liquid	Linear choice (lowest cost), mostly high substitutability, expansion and decline constraints, system integration constraints
POLES	CHP, hydrogen, fuel to liquid	Logit choice model, mostly high substitutability, expansion and decline constraints
REMIND	CHP, Heat pumps, hydrogen (from fossil fuels and biomass with and w/o CCS; electrolytic hydrogen), fuel to gas, fuel to liquid (from fossil fuels and biomass with and w/o CCS), heat plants	Discrete technology choices with high to full substitutability, expansion and decline constraints, system integration constraints
WITCH	None	No discrete technology choices, mostly high substitutability, expansion and decline constraints, system integration constraints

On the same page it is stated:

“In contrast, the other low carbon technologies studied (wind and solar; nuclear energy), as well as a further reduction in energy intensity, have ambiguous impacts. Some IAMs suggest that they would reduce the amount of stranded assets. This is possible because they reduce the carbon intensity of the electricity supply, allowing for more electricity to be generated from fossil fuels. Other IAMs suggest that the development of these technologies would in fact increase the amount of stranded assets. This could be the case as wind, solar and nuclear energy are competing with fossil fuels for meeting electricity demand.”

To a reader it does not become clear what was the objective function in the IAMs. Surely renewable and nuclear power can replace fossil fuel power plants. To what extent the plants become stranded assets depend on their assumed economic lifetime. Should be clarified in paper how these parameters are treated.

Authors’ response: In the revised manuscript, we try to provide as much detail on the IAMs as possible with the above-mentioned summary table in the Methods chapter, presenting the general characteristics, socioeconomic drivers and key technology assumptions of these IAMs. Furthermore, the impact of our

assumptions for the expected lifetime of power generators are described in detail in the Methods section (Pages 29-31 and 38, reproduced below).

“

Lifetime of generators. *Since we are interested in total future electricity generation until the end of the century, we need to make assumptions about when currently operating and in-the-pipeline generators will retire. Only about 5% of power generators in our dataset contain information on when the generator is expected to be retired. For the other ones, we simulate their lifetime based on the retirement year of similar generators. We consider that generators are of same type if they use the same fuel, unit technology type and steam type (usually either subcritical, supercritical, and ultra-supercritical steam conditions), are in the same capacity range (under 400 MW, from 400 to 500 MW, from 500 to 700 MW, from 700 to 900 MW and beyond 900 MW), and started to operate in the same year. We use the most disaggregated levels of technology types available. For example, coal-fired power plants include coal, coke, syngas from gasified coal, synthetic gas from petroleum coke, coke-oven gas, coal steam gas, coal-water slurry, corex process offgas, etc. Then, we assume that the lifetime of power generators of a given type to follow a Poisson distribution with mean value equal to the average known lifetime of generators of same type.*

[...]

We compare our assumptions of the lifetime of generators with the assumptions used previously in the literature in Table AT5. Column (1) shows the average lifetimes of operating and pipeline fossil fuel power generators, which are assumed in our baseline. Columns (2) to (4) list the lifetime assumed in key literatures.

With these baseline assumptions for lifetime, we provide estimates for the global yearly electricity generation levels in Fig. 1. We further show the sensitivity of our estimates to the assumptions of lifetime in Fig. AF5. In Fig. AF5, the “extend” case assumes all fossil fuel power generators extend their lifetime by 10% compared to what we assumed in our baseline estimations while, in the “reduce” case, the expected lifetime is reduced by 10%.

Table AT5: Comparison of lifetime assumptions (Years)

Fuel	Average lifetime assumed in our baseline	Average lifetime in Tong et al. (2019)	Technical lifetime in IRENA (2017)	Economic lifetime in IRENA (2017)
Coal	40	36	50	25
Gas	39	37	30	15
Oil	39	34	50	25

Fig. AF5: Sensitivity of estimated electricity generation (a) and asset stranding (b) to utilisation rates and asset lifetime. All cases show the total estimated electricity generation and asset stranding of operating and in-the-pipeline fossil fuel power plants (coal-, gas- and oil-fired) from 2021 to 2100 in the “all technologies deployed” scenarios, without any conversion. The circles for panel (b) provide the mean across 7 IAMs (GCAM, IMACLIM, IMAGE, MESSAGE, POLES, REMIND, and WITCH). The standard deviation reported in parentheses accounts for the variation across models. The baseline assumption based results presented in the main results are emphasised with the “x” symbol.

Fig. AF5 shows that higher utilisation rates and longer lifetime assumptions lead to more potential fossil fuel electricity generation in the future and therefore more stranded assets. However, the share of stranded assets is relatively stable, with about 50% of future electricity generation from fossil fuel-fired power plants to be stranded in order to achieve the 450ppm climate target.

Reviewer #2

This paper looks at how converting fossil fuel power plants to low-carbon technologies may lower the risks of stranded assets to achieve global 2C goal. This is a very interesting topic that can be of great interest to both the research and policy communities. However, due to strong assumptions and simplified methodology, results of this study tend to overestimate the potential of plant conversion as an approach to lower stranded assets, and the conclusions can be misleading.

Most importantly, strong assumptions are made regarding the potential of fossil plant conversions, which is estimated at the highly aggregate level. It is highly uncertain how many existing and planned plants can be converted. For example, feasibility of CCS plants depends on carbon storage capacity, and feasibility of biomass plants depend on access to biomass resources. Decision making of plant conversion also largely depends on economic viability. How is the retrofitting cost taken into account? Does it make economic sense to retrofit existing plants to lower stranded assets? The generic conversion rate between 0% to 100% does not provide meaningful implication, and more granular analysis is needed to be able to answer the research question.

Authors' response: Thank you for these important comments. We have completely revised the section on plant conversions. Above all, we now assess technical feasibility in the following way:

(a) We could match a plant-level assessment for CCS conversion suitability by Caldecott et al. (2016b) to our own power plant data and we observe that around 24 percent of fossil fuel power units are suitable for CCS conversion. We use this information as a benchmark and then provide results on CCS conversion.

(b) Coal-to-gas conversion has been undertaken in about 80 coal-fired power plants in the US since 2011 (EIA 2020), i.e. in about 5 percent of coal-fired power capacity. We consider that it could likewise be implemented in coal-fired plants located in countries that also have operating gas-fired power plants, and therefore have access to gas infrastructure and supplies. We then assume that between 5 percent and 20 percent of them could be converted. The 5 percent corresponds to the US conversion percentage between 2011 and 2019. 20 percent is an optimistic upper bound equivalent to four times this conversion percentage (note that conversions may accelerate and could happen over a longer time frame than only 9 years).

(c) References show that the co-firing of biomass in coal-fired power plants is currently feasible at 20 percent and technically achievable at 50 percent. Besides technical feasibility, biomass availability is an important limiting factor for conversion. The majority of models used in our analysis are within the estimates of global sustainable biomass potential of 100 EJ per year (Creutzig et al. 2015), except GCAM. We have modified the possible range for plant conversion to consider levels of co-firing in the range of 20-50 percent. We analyze coal to biomass jointly with CCS conversions since most biomass is expected to be used in plants equipped with CCS.

When it comes to economic viability, we unfortunately do not have enough reliable information to predict the costs of these conversions. However, we are able to rely on the hypotheses of the IAMs (which include considerations about costs) to limit the deployment of CCS, gas and biomass according to the forecasts made in the IAMs. Conversions are limited in our projections by the limits included in the IAMs. Since we look at the impact of conversions on stranded assets, we implicitly assume that it is less costly to convert existing assets at risk of stranding than it is to build new ones.

Moreover, the new methods provide lower estimates of the impact of plant conversions on asset stranding, especially since we no longer provide very high upper bounds for the potential of conversions.

The new section on plant conversions on Pages 7-10 reads as follow:

“

Impact of plant conversions

Our analysis above suggests high levels of stranded assets under all IAMs apart from GCAM. These estimates may be overly restrictive, however, because they do not consider that power plants could adapt to the risk of stranded assets by converting to use less carbon-intensive technologies. In Fig. 3 we allow for power plants to be converted and estimate the impact that this would have on stranded assets (compared to Fig. 2). We consider three conversion options: 1) conversion from coal to gas; 2) the installation of CCS in existing plants and in those currently in the pipeline; and 3) the use of biomass in coal-fired power plants.

Coal-to-gas conversion is technically feasible and has been undertaken in more than 80 coal-fired power plants in the US between 2011 and 2019 (EIA, 2020), representing about 5 percent of total US coal-fired power capacity. In Fig 3.a, we estimate the change of stranded assets when considering potential conversions from coal to gas. We consider that coal-fired power plants located in countries that also include operating gas-fired power plants are suitable for conversion since they have access to gas infrastructure and supplies. We then assume that between 5 percent and 20 percent of them could be converted. The 5 percent corresponds to the US conversion percentage between 2011 and 2019. 20 percent is an optimistic upper bound equivalent to four times this conversion percentage (note that conversions may accelerate and could happen over a longer time frame than only 9 years). We furthermore limit possible plant conversions so that the amount of electricity produced from converted plants remains equal or below the electricity produced from gas in the IAMs. Beforehand, we have subtracted the amount already covered by existing gas-fired power plants. Results in Fig. 3.a suggest that coal-to-gas conversions may reduce stranded generation by 10 PWh to 30 PWh on average across all models.

In Fig 3.b, we look at CCS adoption from operating and in-the-pipeline fossil fuel power plants. Caldecott et al. (2016b) assesses that existing generation units are likely to be suitable to CCS installation if they have a capacity above 100MW, are less than 20 years old, emit less than 1000g CO₂/kWh and are located within 40km of geological areas suitable for CCS. Using these criteria, we find that around 24 percent of operating and in-the-pipeline fossil fuel power units would be suitable for CCS conversion. In Fig. 3.b, we show the impact of CCS conversion on stranded assets if either half of those suitable capacities, or all of them, would be equipped with CCS. We furthermore make sure that CCS installation in existing and in-the-pipeline power plants coincides with the take-up of CCS in the IAMs. Therefore, we bound our estimates of electricity generated from converted plants with CCS to be equal or inferior to the total volume of electricity generated from fossil fuels with CCS in the IAMs. The results displayed in Fig. 3.b imply that stranded generation can be reduced by 33 PWh to 52 PWh thanks to the future installation of CCS in operating and in-the-pipeline plants. Reductions are most pronounced in the IAMs in which we found the largest amounts of stranded assets (i.e. POLES, IMACLIM and WITCH).

In Fig. 3.c and 3.d, we consider that coal-fired power plants (with and without CCS) could be adapted to co-fire biomass. So far, more than 150 power plants have fired coal along with biomass, with the majority located in northern Europe and the US (Fernando, 2012). However, in the IAMs, 95 percent of the electricity generated with biomass would come from plants equipped with CCS in the future. Our calculations therefore have to consider the co-firing of biomass in some plants that are not equipped with CCS, and the co-firing of biomass in plants that would be retrofitted with CCS. Fig. 3.c. and 3.d. consider three scenarios of CCS retrofits: no plants are converted to CCS; 50 percent of CCS suitable capacities are converted to CCS; and all CCS suitable capacities are converted to CCS. The last two assumptions are the same as in Fig. 3.b. Furthermore, we allow coal plants to co-fire with biomass. Co-firing ratios can vary. The IEA and IRENA (2013) estimates that a 20 percent co-firing ratio is feasible in most cases, while a 50 percent co-firing ratio is technically achievable. Fig. 3.c. assumes a co-firing ratio of 20 percent and Fig. 3.d a co-firing ratio of 50 percent.

In both figures, the coal-to-biomass potential is restricted to the availability of sustainable biomass feedstock within each IAM. Co-firing without CCS cannot exceed the electricity generated from biomass without CCS in the IAMs, after accounting for the electricity that would already be generated from existing biomass plants. Co-firing with CCS cannot exceed the electricity generated from biomass with CCS in the IAMs. We furthermore respect the co-firing ratios. Therefore, biomass co-firing has to be equal or inferior to either 20 percent or 50 percent of the total of energy generated from converted plants: there cannot be more biomass co-fired than the amount of coal allows for.

In the absence of any CCS conversion, Fig. 3.c suggests that biomass co-firing could reduce the amount of stranded assets by 9 PWh when assuming 20 percent co-firing. With the installation of CCS and 20 percent co-firing, asset stranding could be reduced by 42 to 62 PWh, depending on whether half or all CCS suitable plants would be converted to CCS. The contribution of biomass co-firing is relatively small, at between 9 and 10 PWh, since the installation of CCS alone already allows for a reduction in stranded assets by 33 PWh to 52 PWh (as per Fig. 3.b). In Fig 3.d, with a 50 percent co-firing ratio and no CCS conversion, asset stranding could be reduced by 13 PWh. With a 50 percent co-firing ratio and CCS conversions, asset stranding could be reduced by 47 to 68 PWh in total, of which 14 to 16 PWh would come from biomass co-firing.

Therefore, we find that coal to gas conversions may reduce asset stranding by 10 to 30 PWh, and CCS and biomass together by 33 to 68 PWh. However, the estimates for coal to gas and the other conversion options (of Fig. 3.b to 3.d) have been obtained separately and cannot be directly added up (see the Methods section for more details). Furthermore, there are differences in impact across models that arise from model settings on energy conversion technologies, energy technology choices, substitutability and deployment (see Appendix Table AT2 for more details).

Fig. 3: Impact of plant conversions on global asset stranding. Scatter points represent the estimation from individual models and bars show the model mean. The conversion percentages of Figure 3.a correspond to the share of coal-fired power plants converted to gas. Those of Fig. 3.b to 3.d correspond

to the share of CCS suitable plants converted to CCS. Biomass co-firing ratios vary from 0 in Fig. 3.b to 50 percent in Fig. 3.d.
”

In general, an important message of this paper is that, even assuming a generous potential for plant conversions and optimistic technology assumptions, that there could still be significant amounts of stranded assets.

Second, the definition of asserting stranding is misleading. Stranded assets are mostly relevant to capacity, instead of generation. Lower utilization than today does not necessarily indicate assets stranding. Instead, the authors should look at premature retirement.

Authors’ response: We have relied on the definitions of Caldecott et al (2016a) and Pfeiffer, A. et al. (2018):

“Stranded assets are defined as assets that have suffered from unanticipated or premature write-downs, devaluation or conversion to liabilities.” Caldecott et al (2016a)

“Fossil-fuel generation capacity would have to be stranded, that is, prematurely decommissioned and underutilized, if humanity is to meet the climate goals set out in the Paris Agreement.” Pfeiffer, A. et al. (2018)

In this revised version, we clarify which definition of stranding we use at the onset (see Page 2):

“This paper assesses the potential for new technologies and plant conversions to reduce the need for assets to be “stranded”, i.e. underused or retired early to ensure that global warming remains below 2°C (Caldecott et al., 2016a). We follow Pfeiffer et al. (2018) and define stranded assets in the power sector in terms of stranded generation: the reduction in electricity generation from fossil fuels arising from the necessary underuse and early decommission of power plants consistent with a 2°C scenario.”

Stranding captures the difference between the expected utilization of fossil fuel generation capacity under today’s policy and the actual utilization to comply with climate constraints. There is a link between stranded capacity (i.e. premature retirement of assets) and stranded generation. We chose “stranded generation” because it is inclusive of all forms of asset stranding and presents the number of years of reduced capacity. Calculations of stranded capacity would require making assumptions about which plant retires, where, and when. Intertemporal considerations are difficult since it is possible to reach the same outcome with early retirement of one plant, for example, or the late retirement of two plants.

We add the following explanation in the Methods section (Page 31):

“We define stranding in terms of generation since it captures the overall impact that climate constraints could have on the amount of electricity generation that will have to be cancelled.”

References:

Caldecott, B. et al. (2016a) Stranded Assets: The Transition to a Low Carbon Economy, Lloyd’s of London Emerging Risk Report.

Pfeiffer, A. et al. (2018) ‘Committed emissions from existing and planned power plants and asset stranding required to meet the Paris Agreement’, Environmental Research Letters, 13(5), pp. 054–019. doi: 10.1088/1748-9326/aabc5f.

Third, Fig.1 shows the committed generation from existing plants are much lower than the 2C pathway, and it actually also allows new plants to be added. Showing the 2C pathway of total fossil plants may

create confusion. Literature already suggests any new addition of coal plants is inconsistent with global 1.5/2c climate goals. It is important to show the pathways separately for each plant fuel type.

Authors' response: We clarified that the pathway on Fig. 1 is only illustrative. In fact, it corresponds to an optimistic pathway.

We have modified the text description on Page 4:

"An illustration is provided in Fig. 1. The black dashed lines correspond to one of the scenarios available in the AMPERE Working Package 2 (WP2) database. This scenario assumes that all low-carbon technologies available in the model can be deployed. In this case electricity generation from currently operating plants is roughly within the example's modelled boundary. Nevertheless, adding the electricity generation from in-the-pipeline plants exceeds the scenario's modelled boundary, signalling the asset stranding risk. Other pathways that do not assume availability of all technologies may lead to higher levels of stranded assets."

We have also modified the caption below the Fig. 1 on Page 4:

"The black dashed line is an example of the electricity that could be produced based on one scenario from one IAM, specifically the 450ppm, "all technologies deployed" scenario obtained from the MESSAGE model."

We also added the following text to refer to separate pathways (Page 4):

"Furthermore, stranding risk varies by fuel type, with asset stranding risk for coal-fired power plants much stronger than for other fuels. Separate pathways by fuel type for the same example are provided in Appendix Fig. AF4."

We have added the corresponding Appendix on Page 37:

*"**Separate pathways of electricity generation by fuel.** Fig. AF4 displays the committed electricity generation between 2021 and 2100 by fuel. We also display the electricity generation forecast in the "all technologies deployed" scenario modelled by MESSAGE model for a climate stabilisation at 450ppm. In this example, we observe that coal and oil assets are at high risk of stranding, while gas assets are at lower risk of stranding. Please note that this is only one example for one IAM and one scenario. Results from different models and scenarios would differ significantly."*

Fig. AF4: Breakdown of committed electricity generation between 2021 and 2100 for coal (a), gas (b) and oil (c) separately. Estimates of the amount of future electricity that can be generated from currently operating and in-the-pipeline power plants by fuel. Power units are assumed to be operating at the same utilisation rates as those forecasted in the IEA Stated Policy Scenarios until the end of their expected lifetime. Darker shading indicates the expected generation from currently operating plants, while lighter shading indicates the expected generation from in-the-pipeline plants. The black dashed line is an example for the electricity that could be produced based on one scenario from one IAM. The specific example is the 450ppm, “all technologies deployed” scenario obtained from the MESSAGE model.”

Fourth, the authors should look at 1.5C scenarios to be more relevant with ongoing policy discussion. Only looking at the 2c target may generate misleading messages. The authors may also consider using IAM results from more recent multi-model comparison projects. The AMPERE results seem to be a little bit outdated.

Authors’ response: In this revised version, we provide estimates of asset stranding using the IPCC SR1.5 database in addition to the estimates of the AMPERE database. We refer to these estimates on Page 7:

“We have compared our AMPERE-derived results to those based on the IPCC SR1.5 database (Huppmann et al. 2018). The IPCC SR1.5 database includes 1.5°C as well as 2°C scenarios. We estimate the amount of stranded assets with this database in Appendix Figure AF7. The average amount of stranded assets across all models in 2°C scenarios of IPCC SR1.5 is higher by 12 percent compared to the amount in Fig.2. This is in part because AMPERE scenarios in Fig. 2 assume that all technologies capable of reducing asset stranding have been fully deployed, while 2°C scenarios of IPCC SR1.5 have varying assumptions on technology development. The average amount of stranded assets across all models in the 1.5°C scenarios of IPCC SR1.5 is about 17 percent higher than in the 2°C scenarios of IPCC SR1.5.”

References:

Huppmann, D. et al. (2018) ‘Scenario analysis notebooks for the IPCC Special Report on Global Warming of 1.5°C’. doi: 10.22022/SR15/08-2018.15428.

The estimates using the IPCC SR1.5 database are provided in Supplementary Results on Pages 39-40:

“Estimates of stranded assets using the IPCC SR1.5 database. Our results are obtained by using AMPERE scenarios. Other databases of IAMs could be used to calculate stranded assets with our methodology. We show the results of stranded assets using IPCC SR1.5 database in Fig. AF7. The average amount of stranded assets in 2°C scenarios of IPCC SR1.5 database is about 300 PWh, which is 12 percent higher than the amount in AMPERE 2°C scenarios of Fig. 2. This is mostly because Fig. 2 assumes that all technologies reducing the level of stranded assets are fully deployed, while 2°C scenarios of IPCC SR1.5 have varying assumptions on technology development. The amount of stranded assets under 1.5°C scenarios is higher by about 17 percent compared to 2°C scenarios in IPCC SR1.5 databases. Besides, the order of magnitude, regional distribution and model features of stranded assets are very similar. For example, the WITCH model shows the highest amount of stranded assets, while GCAM model has the lowest amount.
”

Fig. AF7: Estimated stranded assets using the IPCC SR1.5 scenarios (2021-2100). Scatter points represent the mean value of estimations from individual IAMs and bars show the mean value across all models.”

For the core of the text, we preferred to use AMPERE over IPCC SR1.5. This is because, while we can estimate stranded assets with both IPCC SR1.5 and AMPERE, we cannot look at the impact that technology availability would have on stranded assets with IPCC SR1.5. We can only use AMPERE for the analysis of technology options because no other model comparison project provides pairwise scenarios of technology availabilities like AMPERE (as illustrated in Appendix Table AT3, reproduced below).

We explain this on Page 7:

“The rest of this paper is based on the scenarios available in AMPERE, because they allow for the estimation of the effect of technology availability on asset stranding through the pairwise comparison of scenarios with and without a technology (as illustrated in Appendix Table AT3).”

For your convenience, we reproduce below the Appendix Table AT3 with the technology settings in the AMPERE scenarios (on Page 23):

“

Table AT3: Technology settings in AMPERE scenarios.

Technology dimensions	All technologies deployed scenarios	With one technology insufficiently deployed scenarios
Carbon Capture and Storage	CCS is fully available.	CCS never becomes available, including for both fossil fuel and bio-based plants.
Nuclear	Nuclear energy is fully available.	No new investments into nuclear power after 2020; existing plants are fully phased out over their lifetime.
Solar and wind	Advanced* techno-economic assumptions for solar and wind technologies.	Limited contribution of solar and wind to 20% of total power generation, reflecting potential implementation barriers of renewable energy at high penetration rates.
Bioenergy	Total global bio-energy supply shall top out at the level generated endogenously by each model.	Total global bio-energy supply for all sectors from purpose-grown crops, residues and municipal solid waste shall be limited to 100 EJ/ year as primary energy.
Other setting		
Energy intensity	Energy intensity improves at historical rates.	A combination of efficiency measures and behavioural changes leads roughly to a 50% increase of the energy intensity improvement rate compared to historical rates.

Source: AMPERE Work Package 2 model comparison study protocol and Riahi et al. (2015). *: It is left to the modeler's choice what is being considered "advanced". Therefore, the definition is different across IAMs.

”

Reviewer #3

Review of “Can plant conversions and abatement technologies prevent asset stranding in the power sector?”

Key results: This paper estimates the stranded assets in the fossil fuel power plant sector after adjusting for future technologies that might mitigate GHG emissions from those plants. These adjustments create the best-case scenario of for power plant utilization through 2100. The estimates show that even with full deployment of technological advances (some of which are not commercially viable at this time) about 50% of existing and future capacity in the sector will be stranded.

This is an important topic because decarbonization needs to be done in the most cost-effective way possible. Having the best possible estimates of asset stranding will help design the best decarbonization pathways. Moreover, these results are important to financiers when considering funding power plant construction. Banks increasingly face pressure to not finance new power plant construction, but evidence that shows the extent of stranded asset risk may be more persuasive than calls to join the fight against climate change.

Finally, the authors additions of plant conversions from coal-to-gas and then to BECCS help reconcile the highly different results about stranded assets from these models.

Validity: The technical analysis seems sound.

Originality and significance: This is the first paper I am aware of that carefully examines the potential effect of technology on stranded asset risk. As mentioned above, for that reason it is a significant contribution to the decarbonization conversation and should help inform policy decisions.

Data & methodology: The data and approach seem sound and the authors carefully explain the possible limitations of their data.

Appropriate use of statistics and treatment of uncertainties: NA

Conclusions: The conclusions and data interpretation are robust and valid.

Suggested improvements: There are two areas that need to be addressed.

First, the authors analysis is based on seven IAMs with four technologies “fully deployed.” The IAMs give very different results about stranded assets. On page 5 in a paragraph that begins, “The amount of Stranded assets ... ,” the authors explain that this has to do with the structure and assumptions of the models. Later, on page 8 at the end of a paragraph that begins, “In Fig. 4, ... “, the authors state that the impact of CCS can only be computed for the GCAM, REMIND and MESSAGE models. This caveat is mentioned again on page 10, in a paragraph that begins, Furthermore, our estimates” It appears that across the IAMs “fully deployed” means very different things. To provide clarity for readers it would be very helpful to create a table that shows the sorts of technologies each IAMs supports and possibly the time-frame and rate of innovation of these models. A final comment about each model regarding its approach to technological innovation, e.g., endogenous/exogenous, and how diffusion and/or rates of improvement are incorporated in the model, would help readers better understand the variability across the IAM results.

Authors’ response: We thank the reviewer for this suggestion. We have added a summary table in the Data description, presenting the general characteristics, socioeconomic drivers and key technology assumptions of these IAMs (on Pages 19-22):

“Table AT2 presents the general characteristics (panel a), socioeconomic drivers (panel b) and key energy technology assumptions (panel c) of these IAMs. The information is compiled from IPCC (2014) and Kriegler et al.(2015). As a modelling comparison project, AMPERE harmonized GDP and population assumptions across models to facilitate the analysis of model differences. Nevertheless,

models differ in numerous ways and lead to varying results, as showed in our results. Additional details on the models used in AMPERE project are provided in Kriegler et al. (2015).”

Table AT2: Overview of IAM models used in this study

a) General characteristics					
Model name	Institution	Reference	Equilibrium type	Modelling approach	Time horizon
GCAM	JGCRI, US	JGCRI (2010)	Partial	Recursive dynamic	2100
IMACLIM	CIREC and SMASH, France	Sassi et al. (2010)	General	Recursive dynamic	2100
IMAGE	UU and PBL, Netherlands	Bouwman et al. (2006)	Partial	Recursive dynamic	2100
MESSAGE	IIASA, Austria	Messner et al. (2000)	General	Intertemporal optimization	2100
POLES	EC-JRC, Belgium	IPTS (2010)	Partial	Recursive dynamic	2100
REMIND	PIK, Germany	Leimbach et al. (2010)	General	Intertemporal optimization	2100
WITCH	FEEM and CMCC, Italy	De Cian et al. (2009)	General	Intertemporal optimization	2100
b) Social economic drivers					
Model name	Exogenous drivers		Endogenous drivers		
GCAM	Population, GDP, labour participation rate, labour productivity		None		
IMACLIM	Labour productivity, energy technical progress, population, active population		None		
IMAGE	Exogenous GDP, GDP per capita, population		Energy demand, renewable price, fossil fuel prices, carbon prices, technology progress, energy intensity, preferences, learning by doing, agricultural demand, value added		
MESSAGE	Labour productivity, energy technical progress, GDP per capita, population		None		
POLES	Exogenous GDP, population		Value added, mobility needs, fossil fuel prices, buildings surfaces		
REMIND	Labour productivity, energy efficiency parameters of the production function, population		Investments in industrial capital stock and specific energy technology capital stocks, endogenous learning-by-doing for wind and solar power as well as electric and fuel cell vehicle technologies (global learning curve, internalized spillovers).		
WITCH	Total factor productivity, labour productivity, capital technical progress		None		
c) Key energy technology assumptions					
Model name	Energy Conversion technologies		Energy technology choice, substitutability and deployment		

GCAM	CHP, hydrogen from coal, oil, gas, and biomass, w/o and w/ CCS, nuclear and solar thermochemical, fuel to gas, coal to gas w/o CCS, biomass (w/o and w/ CCS), fuel to liquid, coal to liquids (w/o and w/ CCS), gas to liquids (w/o and w/ CCS), biomass to liquids (w/o and w/ CCS)	Logit choice model with usually high substitutability
IMACLIM	Fuel to liquid	Logit choice model, mostly high substitutability in some sectors and mostly low substitutability in other sectors, expansion and decline constraints, system integration constraints
IMAGE	CHP, hydrogen	Logit choice model, mostly high substitutability, expansion and decline constraints, system integration constraints
MESSAGE	CHP, hydrogen, fuel to gas, fuel to liquid	Linear choice (lowest cost), mostly high substitutability, expansion and decline constraints, system integration constraints
POLES	CHP, hydrogen, fuel to liquid	Logit choice model, mostly high substitutability, expansion and decline constraints
REMIND	CHP, Heat pumps, hydrogen (from fossil fuels and biomass with and w/o CCS; electrolytic hydrogen), fuel to gas, fuel to liquid (from fossil fuels and biomass with and w/o CCS), heat plants	Discrete technology choices with high to full substitutability, expansion and decline constraints, system integration constraints
WITCH	None	No discrete technology choices, mostly high substitutability, expansion and decline constraints, system integration constraints

The detailed definition of “fully deployed” is also provided in Appendix Table AT3 (on Page 23, reproduced below). We also show the global electricity generation from difference fuels / technologies between 2021 and 2100 in the “all technology deployed” scenarios in Figure AF1 (on Page 25, reproduced below). We have added the following text on Page 5:

“Appendix Table AT3 provides more detail on the definition of the “technologies fully deployed” scenarios compared to other scenarios. In a nutshell, no pre-set constraint is imposed on the function of technology deployment. In other scenarios, limits are usually imposed, whether the potential is limited (e.g. bioenergy limited to 100EJ per year or solar and wind are limited to 20 percent of total power generation) or the technology is not allowed (e.g. CCS is not available and no new investments in nuclear power after 2020). The “all technologies deployed” scenarios also assume that energy intensity continues to reduce every year at historical rates. Appendix Figure AF1 displays the global electricity generation by technology in the “all technologies deployed” scenarios of each IAM.”

Table AT3: Technology settings in AMPERE scenarios.

Technology dimensions	All technologies deployed scenarios	With one technology insufficiently deployed scenarios
--	--

Carbon Capture and Storage	CCS is fully available.	CCS never becomes available, including for both fossil fuel and bio-based plants.
Nuclear	Nuclear energy is fully available.	No new investments into nuclear power after 2020; existing plants are fully phased out over their lifetime.
Solar and wind	Advanced* techno-economic assumptions for solar and wind technologies.	Limited contribution of solar and wind to 20% of total power generation, reflecting potential implementation barriers of renewable energy at high penetration rates.
Bioenergy	Total global bio-energy supply shall top out at the level generated endogenously by each model.	Total global bio-energy supply for all sectors from purpose-grown crops, residues and municipal solid waste shall be limited to 100 EJ/ year as primary energy.
Other setting		
Energy intensity	Energy intensity improves at historical rates.	A combination of efficiency measures and behavioural changes leads roughly to a 50% increase of the energy intensity improvement rate compared to historical rates.

*Source: AMPERE Work Package 2 model comparison study protocol and Riahi et al. (2015). *: It is left to the modeler's choice what is being considered "advanced". Therefore, the definition is different across IAMs.*

Fig. AF1: Global electricity generation from different technologies between 2021 and 2100 in AMPERE “all technology deployed” scenarios (450ppm optimal scenarios)

Carefully explaining what ‘fully deployed’ means in each model pops up in Figure 3. Figure 3 shows the reduction in asset stranding when power plants are converted from coal to gas and coal to biomass. Several models show almost no reduction in stranded assets suggesting that these conversions are part of the full deployment of technology built into the models.

Authors’ response: The parameter that matters in the IAMs for our calculation of stranded assets is the volume of expected energy generation from each source in each year. The IAMs make different assumptions on the substitutability of different fuels. In all IAMs except GCAM, the possibility of plants being converted is not contemplated to model the electricity generation pathway. In contrast, GCAM accounts for more possibilities of conversions, including from coal to gas and coal to biomass (as shown in Appendix Table AT2, panel c, on Page 21).

While this may have an impact on the electricity generation pathway in GCAM versus the other models, it is difficult to determine the separate impact of this assumption as other assumptions in the models may also create differences in estimates. When GCAM contemplates the possibility of conversions, what the model will do is to calculate amounts of coal, gas and biomass that account for the possibility of plant conversions later. It may allow for more coal at the beginning of the period, if generation is then converted into gas or biomass. This would reduce asset stranding. Having said so, due to the heterogeneity of assumptions on technology development across models, we believe the best that can be done is to rely on a set of different IAMs to obtain robust results.

We explain this technical limitation in our Data section (Page 22):

“Appendix Table AT2, panel c, shows that plant conversions from coal to gas or coal to biomass are not included in the calculations of electricity generation by fuels in the IAMs, except for GCAM. Probably, this implies that GCAM might allow for more coal at the beginning of the century, since generation can be converted into gas or biomass. However, it is difficult to determine the separate impact of this assumption, since other assumptions in the models may also create differences in electricity generation levels by fuels in the IAMs. Due to the heterogeneity of assumptions on technology development, it is essential to rely on a set of different IAMs to obtain robust results.”

My second concern or preference is also related to the deployment inherent in the IAMs and thereby in the results. Can the authors somehow link their results to well-known forecasts or energy scenarios such as the IEA’s assumptions about CCS or CCUS in its models. For example, in the IEA’s CCUS in Clean Energy Transitions, September 2020 (<https://www.iea.org/reports/ccus-in-clean-energy-transitions/ccus-in-the-transition-to-net-zero-emissions>) a table shows cumulative CO₂ capture from the power generation sector of 87,529 Mt. Is there any way to relate this assumption of the IEA SDS to the results the authors are presenting? My question is, How does the IEA assumption compare to the “fully deployed” assumptions of the IAMs used in this study?

Authors’ response: Thank you for pointing this out. We compare CO₂ capture in our scenarios with numbers in IEA SDS scenarios in Appendix Fig. AF2 (on Page 26). We refer to this on Pages 10-11:

“Several scenarios within and outside the AMPERE database make less optimistic assumptions about technology take-up. For example, the IEA’s Sustainable Development Scenario assumes a late take-up of biomass with CCS (see Appendix Fig. AF2). Between 2021 to 2050 (when most asset stranding would happen for existing and in-the-pipeline plants in our database), we can calculate that the IEA scenario corresponds to a total energy production from biomass with CCS of only 4 PWh. In the “all technology deployed” scenarios of AMPERE, this technology is being deployed at scale. Therefore, electricity generation from biomass with CCS between 2021 and 2050 would correspond to 40 PWh in these scenarios on average across all IAMs.”

The added text and figure on Page 26 are reproduced below:

“We compare the CO₂ captured in the AMPERE “all technology deployed” scenarios with the IEA’s Sustainable Development Scenario (IEA, 2020) in Fig. AF2. The AMPERE scenarios and the IEA’s Sustainable Development Scenario make relatively similar assumptions for CCS take-up from all sources except biomass. However, assumptions regarding biomass with CCS are very different. The IEA’s Sustainable Development Scenario assumes a late take-up of biomass with CCS, while in the “all technology deployed” scenarios of AMPERE, this technology is being deployed at scale. In the section on “Impact of energy demand, alternative electricity sources, CCS and bioenergy availability”, we discuss the possible impact of low technology diffusion on the amount of stranded assets.”

Fig. AF2: CO₂ capture from biomass and other sources in the AMPERE “all technology deployed” scenarios and the IEA Sustainable Development scenarios (2021-2070). We provide the information separately for CCS from all sources except biomass (a) and CCS from biomass (b) until 2070. The IEA scenarios only model pathways until 2070.

My final comment is that the authors seem to under-sell what they have done. On page 10, in a paragraph that begins, Furthermore, our estimates ... ,” there is a sentence that I didn’t understand or notice the first couple of readings of the paper. It says, “Therefore, our analysis reconciles the findings from different models.” If I understand Figures 2 and 3, together they show this reconciliation of results. Figure 3 adds technologies to models in which they were originally under-represented. For example, in Figure 2 the WITCH model predicts stranded assets of about 250 Pw of capacity. But in Figure 3, with plant conversions and CCS, this model shows a reduction of stranded assets of about 180 Pw at a 100% conversion ratio. I think it would be very powerful to combine these results and show the convergence of the models. Of course, a 100% conversion to BECCS is unlikely because of the lack of CCS commercialization and penetration, the lack of biomass fuels in some regions, and financial constraints, which are the ‘technical and economic barriers’ the authors mention on page 10. The authors could then discuss asset stranding at lower conversion rates and its implications to future investment in fossil fuel fired power plants.

Authors' response: Thank you for this comment. We added a sentence in introduction on this, and modified the end of our introduction to insist on the robustness of our results across different IAMs.

The updated text on Page 3:

“Furthermore, looking at differences in assumptions regarding technology development allows us to reconcile differences in results across different models of long-term energy generation.

Our results suggest that, even in the presence of a strong industry response to develop low carbon and negative emission technologies or convert current and planned assets to be less carbon-intensive, the expected amount of required stranding would remain substantial. These results are robust across commonly-used forecasts of energy generation in the 21st century.”

References: The manuscript references are thorough and appropriate.

Clarity and context: The abstract's last sentence is a bit confusing.

The sentence reads, “Fossil fuel plants remain at substantial risk of stranding under optimistic assumptions, and even more so if CCS and bioenergy are deployed at scale.” I am not sure what “at scale” refers to. Maybe change this to something related to “at current rates of innovation” or “estimated rates of technological penetration”?

Authors' response: We thank you for spotting this typo. We amended the sentence with “*if CCS and bioenergy are not deployed at scale.*”

There is a small typo on page 8 in the paragraph that begins, “Fig 4 shows” “The last sentences includes “ ... to the atmosphere as the it stores the .. , which has an extra word.

Authors' response: Thank you for pointing these out. We have corrected these typos in the revised manuscript.

REVIEWER COMMENTS

Reviewer #1 (Remarks to the Author):

The authors have made a thorough revision of the paper to meet both my general and my detailed comments. They have now provided a clearer definition on the concept of stranded assets. As for the assumptions on how coal power plants can be converted/replaced I think this is still rather academic and giving the impression to be made without much knowledge on power plant technology but considering the high “system level” of the analysis it may suffice. The reference to statistics of coal to gas power plant conversions in the US is ok, but it is still not convincing to what extent this is relevant on a global scale and what is actually meant by the plant conversion (what is replaced on the power plant island?). Access to low-cost shale gas has been quite unique for the US and to what extent similar development can be expected in other countries is not known (will depend on natural gas markets). Since the analysis lacks detail in these respects one could question if it is fair to claim that “this paper accounts for the possible response of the industry towards developing technological solutions when facing the risk of assets being stranded”. This will require a more refined analysis. However, if understood that this is from a global overall perspective it may still be valid and, thus, I would recommend authors to moderate the above formulation to reflect this. It should be up to the editor to decide if assumptions around coal to gas conversions should be further strengthened.

Reviewer #2 (Remarks to the Author):

Overall, the authors did a great job addressing reviewers’ comments, especially by updating the approach to identify the conversion potential for plants and adding the comparison to IPCC 1.5C scenarios, the manuscript has been improved. Here are some additional comments for the authors’ further consideration.

First, I would suggest using stranded generation instead of stranded assets throughout the paper to more accurately describe what’s being quantified. Also, in the abstract, I find the phrase “assets capable of producing 267 PWh electricity” misleading. Should simply change it to “total stranded generation”.

Second, the authors should further clarify between the two sets of technology scenarios: plant conversions vs. mitigation technology availability. In particular, CCS retrofit vs. CCS and biomass co-firing vs. bioenergy, are similar and sometimes may create confusion. When writing the key findings (i.e. in the abstract), should make it clear these are two separate analyses.

Third, the sensitivity analysis on lifetime and utilization assumptions is very useful, but I find the conclusion unclear: “Higher utilisation rates and longer lifetime assumptions would lead to more future electricity generation and therefore more asset stranding. However, the share of assets that would have to be stranded is stable.” Why is that? The assumptions only affect expected generation under BAU, but not the generation under policy. With the same 2C generation pathways, doesn’t the share of stranding increase?

Fig.1 legend is unclear. Clarify it is electricity generation from fossil plants without CCS, same for the dash line?

Reviewer #3 (Remarks to the Author):

The authors have completely addressed my comments as well as addressing the excellent comments of the other reviewers. I don't have any further comments or questions.

The estimates for stranded assets, given the possibility of technological conversion of power plants to lower emitting fuels, seem rigorous and will help inform public policy as we begin to think more strategically about the transition to a low-carbon economy. Moreover, the paper establishes a strong foundation for further work that might examine issues of economic viability of conversion or technological conversion on a more granular level (potential possible with emerging work in spatial finance).

The paper should also inform public models, such as En-Roads, regarding the limitations of conversion.

Overall, this is a very interesting and timely paper and I appreciate being able to participate in its publication.

POINT-BY-POINT RESPONSE TO THE REVIEWERS' COMMENTS

We would like to take this opportunity to express our sincere thanks to the reviewers for their time and extremely helpful comments. The detailed responses below aim to address each comment (authors' responses in black, right after each comment in blue). We also provide a manuscript with changes in highlight. We hope that all three reviewers will be satisfied with the changes.

Reviewer #1

The authors have made a thorough revision of the paper to meet both my general and my detailed comments. They have now provided a clearer definition on the concept of stranded assets. As for the assumptions on how coal power plants can be converted/replaced I think this is still rather academic and giving the impression to be made without much knowledge on power plant technology but considering the high “system level” of the analysis it may suffice. The reference to statistics of coal to gas power plant conversions in the US is ok, but it is still not convincing to what extent this is relevant on a global scale and what is actually meant by the plant conversion (what is replaced on the power plant island?). Access to low-cost shale gas has been quite unique for the US and to what extent similar development can be expected in other countries is not known (will depend on natural gas markets). Since the analysis lacks detail in these respects one could question if it is fair to claim that “this paper accounts for the possible response of the industry towards developing technological solutions when facing the risk of assets being stranded”. This will require a more refined analysis. However, if understood that this is from a global overall perspective it may still be valid and, thus, I would recommend authors to moderate the above formulation to reflect this. It should be up to the editor to decide if assumptions around coal to gas conversions should be further strengthened.

Authors' response: We have revised the section on coal-to-gas conversions to strengthen and clarify our assumptions.

- 1) We modified the paragraph describing coal-to-gas conversion to include a few references on the required processes, as well as references on coal-to-gas conversions outside of the US (see Page 8):

“Coal power plants can be modified to operate only on natural gas (conversion); to fire either coal or natural gas (dual fuel); or to fire both coal and natural gas at the same time (co-firing) (Bartnik, 2013; Carapellucci and Giordano, 2015; Bedilion, 2017; Mills, 2018; Stevick, 2019). Converting a coal-fired boiler to gas requires adding new equipment, such as gas igniters, scanners, piping and valves. It also requires a modification of burner management and combustion control systems, an adjustment of pressure-part through the convection pass and a layup of coal and ash handling equipment (Bedilion, 2017; TransAlta, 2020). So far, coal-to-gas conversion has been undertaken in more than 80 coal-fired power plants in the US between 2011 and 2019 (EIA, 2020), representing about 5 percent of total US coal-fired power capacity. Other countries such as Canada (TransAlta, 2020) and United Kingdom (Staff, 2019) also have several coal-to-gas projects at different stages of development.”

- 2) Our global-scale analysis relies on credible ballpark figures for the likely shares of converted plants in the coming decades.
 - a. We kept the minimum *technical* limit at 5 percent for coal-to-gas conversions, considering that this percentage has been achieved in the last decade in the US. Please note that this only represents a technical limit. In our forecasts, this limit is only achieved if there is enough demand for more electricity to be produced from gas-fired power plants in the IAM, in addition to the electricity that is already covered by existing gas-fired power plants. In the IAMs used to compute Figure 3, when we assume a maximum conversion potential of 5 percent, on average

only 2.8 percent of coal-to-gas conversion occurs due to the limits imposed by IAMs regarding the use of gas in power generation.

- b. We provide a stronger justification for setting a maximum technical limit at 20 percent conversion rate for coal-to-gas conversions. Domeshek and Burtraw (2021) have suggested that a co-firing ratio of 20 percent should be adopted as a standard in the US. In practice, higher co-firing rates are feasible (Stevick, 2019; TransAlta, 2020), even though unusual. Likewise, we use this figure of 20 percent as a technical limit. In our estimations, the conversion potential is further limited by the economic factors accounted for in the IAMs regarding the use of gas in power generation. The 20 percent conversion rate is only achieved when there is enough electricity forecasted to be produced from gas-fired power plants in the IAM, and the electricity demand is not fully covered by existing and pipeline gas-fired power plants. In the IAMs, when we allow for up to 20 percent of conversions of coal-fired electricity to gas-fired, only 10.6 percent on average across the IAMs is converted due to the limits on gas-fired power generation imposed by the IAMs.

We clarify points 2.a and 2.b in the core of the text on Pages 8 and 9:

“We then assume that between 5 percent and 20 percent of them could be converted. The 5 percent corresponds to the US conversion percentage between 2011 and 2019. 20 percent is an optimistic upper bound based on the possibility that gas is co-fired with coal. In that regard, Domeshek and Burtraw (2021) have suggested that US coal-fired power plants could adopt a 20 percent gas co-firing standard. Because these figures of 5 to 20 percent are derived from the US experience, they are likely to constitute upper bounds for what could happen globally. In the US, natural gas is relatively cheap because of the deployment of shale gas and the significant reduction in gas prices that followed. We furthermore limit possible plant conversions so that the amount of electricity produced from converted plants remains equal or below the electricity produced from gas in the IAMs. Beforehand, we have subtracted the amount already covered by existing gas-fired power plants. In this way, our projections for the impact of plant conversions take into account future forecasts for the use of gas in power generation globally.

Results in Fig. 3.a suggest that coal-to-gas conversions may reduce stranded generation by 10 PWh to 30 PWh on average across all models. When we assume a maximum conversion potential of 5 percent of coal-fired power plants, on average 2.8 percent of coal-to-gas conversion occurs in our projections due to the limits imposed by IAMs regarding the use of coal and gas in power generation. When we allow for up to 20 percent of conversions, 10.6 percent of coal-fired generation is converted on average due to the limits imposed by the IAMs.”

- 3) To tone down the affirmation that “This paper accounts for the possible response of the industry towards developing technological solutions when facing the risk of assets being stranded”, we highlight that much uncertainty remains regarding the conversion potential of plants since it will depend on future technology development. This is done in the discussion section on Page 14:

“Also, even though we provide lower- and upper-bound estimates for the reduction in stranded generation from plant conversions, there is much uncertainty regarding the potential and pace of future conversions.”

References:

Bartnik, R. (2013) *The Modernization Potential of Gas Turbines in the Coal-fired Power Industry: Thermal and Economic Effectiveness*. Springer Science & Business Media.

Bedilion, R. (2017) ‘Technology considerations for coal-to-natural gas conversions’, in EIA Energy Conference. Available at: https://www.eia.gov/conference/2017/pdf/presentations/robin_bedellin.pdf.

Carapellucci, R. and Giordano, L. (2015) Upgrading existing coal-fired power plants through heavy-duty and aeroderivative gas turbines, *Applied Energy*, 156, pp. 86–98. doi: 10.1016/j.apenergy.2015.06.064.

Domeshek, Maya, and Dallas Burtraw. 2021. “Reducing Coal Plant Emissions by Cofiring with Natural Gas.” Washington, D.C. <https://www.rff.org/publications/issue-briefs/reducing-coal-plant-emissions-by-cofiring-with-natural-gas/>.

Mills, S. (2018) Combining solar power with coal-fired power plants, or cofiring natural gas, *Clean Energy*, 2(1), pp. 1–9. doi: 10.1093/ce/zky004.

Staff, R. (2019). British Government Approves Drax Gas Conversion Plans at Power Plant. Reuters, October 7, 2019. <https://www.reuters.com/article/uk-britain-drax-group-idUKKBN1WM0SC>.

Stevick, Dale. 2019. “Co-Firing Natural Gas and Coal.” Fossil Consulting Services. 2019. <https://www.fossilconsulting.com/2019/03/27/co-firing-natural-gas-and-coal/>.

TransAlta (2020) Coal-to-Gas Conversions Project. Available at: <https://transalta.com/our-operations/coal-to-gas/> (Accessed: 17 August 2021).

Reviewer #2

Overall, the authors did a great job addressing reviewers’ comments, especially by updating the approach to identify the conversion potential for plants and adding the comparison to IPCC 1.5C scenarios, the manuscript has been improved. Here are some additional comments for the authors’ further consideration.

First, I would suggest using stranded generation instead of stranded assets throughout the paper to more accurately describe what’s being quantified. Also, in the abstract, I find the phrase “assets capable of producing 267 PWh electricity” misleading. Should simply change it to “total stranded generation”.

Authors’ response: Thank you for your suggestion. We have replaced “stranded assets” with “stranded generation” throughout the paper (after introducing the definition of “stranded assets” in the introduction) and updated the abstract accordingly.

Second, the authors should further clarify between the two sets of technology scenarios: plant conversions vs. mitigation technology availability. In particular, CCS retrofit vs. CCS and biomass co-firing vs. bioenergy, are similar and sometimes may create confusion. When writing the key findings (i.e. in the abstract), should make it clear these are two separate analyses.

Authors’ response: Thank you very much for these comments. We have tried to clarify this in the abstract.

The updated text read as follows:

Continued fossil-fuel development puts existing assets at risk of exceeding the capacity compatible with limiting global warming below 2°C. However, it has been argued that plant conversions and new abatement technologies may allow for a smoother transition. We quantify the impact of future technology availability on the need for fossil fuel power plants to be stranded, i.e. decommissioned or underused. Even with carbon capture and storage (CCS) and bioenergy widely deployed in the future, **a total of 267 PWh electricity generation** (ten times global electricity production in 2018) may still be stranded. Coal-to-gas conversions could prevent 10-30 PWh of stranded generation. CCS retrofits, **combined with** biomass co-firing, could prevent 33-68 PWh. In contrast, lack of deployment of CCS

or bioenergy could increase stranding by 69 or 45 percent respectively. Stranding risks remain under optimistic technology assumptions and even more so if CCS and bioenergy are not deployed at scale.

Third, the sensitivity analysis on lifetime and utilization assumptions is very useful, but I find the conclusion unclear: “Higher utilisation rates and longer lifetime assumptions would lead to more future electricity generation and therefore more asset stranding. However, the share of assets that would have to be stranded is stable.” Why is that? The assumptions only affect expected generation under BAU, but not the generation under policy. With the same 2C generation pathways, doesn’t the share of stranding increase?

Authors’ response: Thank you for the clarifying comment. You are correct, the share of stranding increases slightly when lifetimes and utilisation rates increase. For example, in Figure AF5, when keeping baseline lifetime assumptions and increasing utilisation rates from minimum to maximum assumptions, the share of stranding increases from 48% to 49% and 54%. We have clarified the text on Page 6 as follows:

“Higher utilisation rates and longer lifetime assumptions would lead to more future electricity generation and therefore more stranded generation. However, even though the share of stranded generation increases with longer lifetimes and higher utilisation rates, it remains close to about 50 percent of future electricity generation in most cases.”

Fig.1 legend is unclear. Clarify it is electricity generation from fossil plants without CCS, same for the dash line?

Authors’ response: Thank you for the clarifying comment. We have added the following text to the legend of Fig.1 on Page 4:

“All the electricity generation showed in this figure is from fossil fuel plants without CCS.”

Reviewer #3

The authors have completely addressed my comments as well as addressing the excellent comments of the other reviewers. I don't have any further comments or questions.

The estimates for stranded assets, given the possibility of technological conversion of power plants to lower emitting fuels, seem rigorous and will help inform public policy as we begin to think more strategically about the transition to a low-carbon economy. Moreover, the paper establishes a strong foundation for further work that might examine issues of economic viability of conversion or technological conversion on a more granular level (potential possible with emerging work in spatial finance).

The paper should also inform public models, such as En-Roads, regarding the limitations of conversion.

Overall, this is a very interesting and timely paper and I appreciate being able to participate in its publication.

Authors’ response: Thank you very much for these comments. We are very glad that you liked our paper.

REVIEWERS' COMMENTS

Reviewer #1 (Remarks to the Author):

I am satisfied with the actions taken by the authors.

Reviewer #2 (Remarks to the Author):

The authors have addressed all the reviewers' comments. Suggest publication. This is a great contribution to the literature to explore different strategies for accelerated fossil phaseout.

Small notes on Fig.2-4, should change the y-axis title to stranded generation in consistent with the figure legend.

POINT-BY-POINT RESPONSE TO THE REVIEWERS' COMMENTS

We would like to take this opportunity to express our sincere thanks to the reviewers for their time and extremely helpful comments. Please find below our responses to this last round of comments (authors' responses in black, right after each comment in blue).

Reviewer #1

I am satisfied with the actions taken by the authors.

Authors' response: Thank you very much for all the comments. We are very glad that you liked our paper.

Reviewer #2

The authors have addressed all the reviewers' comments. Suggest publication. This is a great contribution to the literature to explore different strategies for accelerated fossil phaseout.

Small notes on Fig.2-4, should change the y-axis title to stranded generation in consistent with the figure legend.

Authors' response: We have modified all the relevant y-axis titles in the figures. Thank you again for all the comments. We are very glad that you liked our paper.